# Genes linked to schistosome resistance identified in a genome-wide association study of African snail vectors

Tom Pennance [1,6], Jacob A. Tennessen [2,6], Johannie M. Spaan [1], Tammie J. McQuistan[1], George Ogara[3], Fredrick Rawago [3], Kennedy Andiego[3], Boaz Mulonga[3], Meredith Odhiambo[3], Martin W. Mutuku[4], Gerald M. Mkoji[4], Eric S. Loker[5], Maurice R. Odiere [3] & Michelle L. Steinauer [1] ✉

Schistosomiasis, a neglected tropical disease, is transmitted by freshwater snails. Interruption of transmission will require novel vector-focused interventions. We performed a genome-wide association study of African snails, *Biomphalaria sudanica*, exposed to *Schistosoma mansoni* in an endemic area of high transmission in Kenya. Two snail genomic regions, *SudRes1* and *SudRes2*, were significantly associated with snail resistance to schistosomes. *SudRes1* includes receptor-like protein tyrosine phosphatases while *SudRes2* includes a class of leucine-rich repeat-containing G-protein coupled receptors, both comprising diverse extracellular binding domains suggestive of host-pathogen interaction. Resistant and susceptible haplotypes show numerous coding differences including presence/absence of entire genes. No loci previously tied to schistosome resistance in a neotropical snail species showed any association with compatibility suggesting that loci involved in the resistance of African vectors are distinct. Snail ancestry was also strongly correlated with parasite compatibility. These results will inform future efforts to predict and manipulate immunity of a major schistosome vector.

Schistosomiasis is a global scourge, taking a large toll on people who have the fewest resources. Affecting over 260 million people, it is the parasitic disease with the greatest impact on health worldwide after malaria[1,2]. Within the last decade, schistosomiasis control program goals have shifted from reduction of morbidity to elimination or interruption of schistosomiasis as a public health problem by 2030[1,3,4]. However, the toolbox with which to combat schistosome transmission has remained virtually the same, dominated by one main approach: mass drug administration (MDA) of praziquantel[5]. It is increasingly recognized that in addition to chemotherapy, successful control and elimination will require targeting the aquatic snails which serve as

intermediate hosts of the schistosome parasites and transmit them to humans[6,7]. Part of the reason MDA alone is insufficient is that even with effective drug treatment, people become rapidly reinfected by infected snails in the environment[8–10]. Schistosomes form chronic infections in snails and continually release hundreds to thousands of infectious stages (cercariae) into the environment daily[11].

Historically, schistosomiasis control programs that are focused on snail control have been the most successful at reducing or eliminating schistosomiasis[12,13]; however, snail-directed control methods are limited and have negative impacts. Molluscicides are indiscriminately toxic and are impractical to apply to vast habitats[12,14]. Furthermore,

[1]College of Osteopathic Medicine of the Pacific – Northwest, Western University of Health, Sciences, Lebanon, OR, USA. [2]Harvard T.H. Chan School of Public Health, Boston, MA, USA. [3]Centre for Global Health Research, Kenya Medical Research Institute (KEMRI), P. O. Box 1578-40100 Kisumu, Kenya. [4]Centre for Biotechnology Research and Development, Kenya Medical Research Institute (KEMRI), P.O. Box 54840–00200 Nairobi, Kenya. [5]Department of Biology, Center for Evolutionary and Theoretical Immunology, Parasite Division Museum of Southwestern Biology, University of New Mexico, Albuquerque, New Mexico, USA. [6]These authors contributed equally: Tom Pennance, Jacob A. Tennessen. ✉e-mail: msteinauer@westernu.edu

snail population rebound post-molluscicide application is predicted to increase schistosome transmission[15].

Given the absence of suitable snail vector control methods for contemporary public health interventions, there is a need for new approaches to be developed[16]. Genomic and transcriptomic data and resources enable approaches like genome-wide association studies (GWAS) that identify genomic regions of snail vectors involved in resisting schistosome infection[17]. Once identified, snail genes and genomic variants associated with resistance to schistosomes could be monitored in wild populations and potentially be manipulated so that snail resistance to schistosomes is enhanced and transmission to humans is interrupted.

The feasibility of manipulating snail resistance to schistosomes may follow similar approaches to engineering resistant hosts for disease control using CRISPR-Cas and associated gene drive technologies[18,19]. However, much of the elegant work regarding transcriptomics and genomics of schistosome-snail compatibility has only addressed these questions in laboratory models of the South American vector of *Schistosoma mansoni*, *Biomphalaria glabrata*[17,20–24]. Very little is known regarding how this body of knowledge will translate to African vectors of *S. mansoni*, through which 90% of *S. mansoni* transmission occurs[25]. The recent publication of two African *Biomphalaria* species genomes and transcriptomes[26,27] provide a path toward molecular-informed snail control in hotspots of transmission.

With the goal of identifying and describing the genetic architecture underlying resistance of African snails to *S. mansoni*, we performed a pooled genome-wide association study (pooled-GWAS)[28] using a wild population of *Biomphalaria sudanica* originating from the shores of Lake Victoria closest to a persistent hotspot of schistosomiasis in Kanyibok, western Kenya[10]. We identify two large-effect loci and a strong influence of ancestry on snail resistance to schistosome infection, defined here as complete parasite clearance (including

parasite DNA) from snail tissue following exposure. These results reveal how immunogenetics and population demographics contribute to vectorial competence in a natural vector population with direct impact on human health.

## Results

### Pooled-GWAS reveals multiple variants strongly enriched in resistant snails

Of 1400 F1 *B. sudanica*, whose parents originated from Anyanga Beach (Lake Victoria, western Kenya; Fig. S1), exposed to eight freshly hatched *S. mansoni* miracidia from local schoolchildren, 1109 snails remained in the GWAS study after excluding 254 that died prior to screening for infection and 37 that yielded insufficient genomic DNA (gDNA) quality. The final sample set comprised 615 and 393 snails that were positive (i.e., releasing *S. mansoni* cercariae) or negative (i.e., not releasing *S. mansoni* cercariae nor PCR positive[29]), respectively. An additional 101 snails that were negative for cercariae but positive for *S. mansoni* gDNA were not considered further. Two equal mass gDNA pools for the pooled-GWAS comprised 493 positive and 295 negative snails (122 positive and 98 negative snails reserved for amplicon panel genotype-validation). The pooled-GWAS sequencing (Illumina paired-end 150 bp, NovaSeq 6000 S4 flow cell) yielded on average 1.5x coverage per snail (Supplementary Data 1). A total of 4,498,972 variants were retained for analysis. Correlation between sequencing technical replicates of positive and negative pooled gDNA was significantly positive (Fig. S2).

In the pooled-GWAS results, genotype-phenotype association $p$ values ranged as low as 1e-30, including 45 variants (0.001%) with $p \le$ 1e-15 and 1930 variants (0.04%) with $p \le$ 2.5e-9 (Fig. 1; Fig. S3). Rather than simply defining a genome-wide significance threshold to identify candidates to be validated, we prioritized genomic regions meeting a dual-variant criterion, whereby two or more proximate

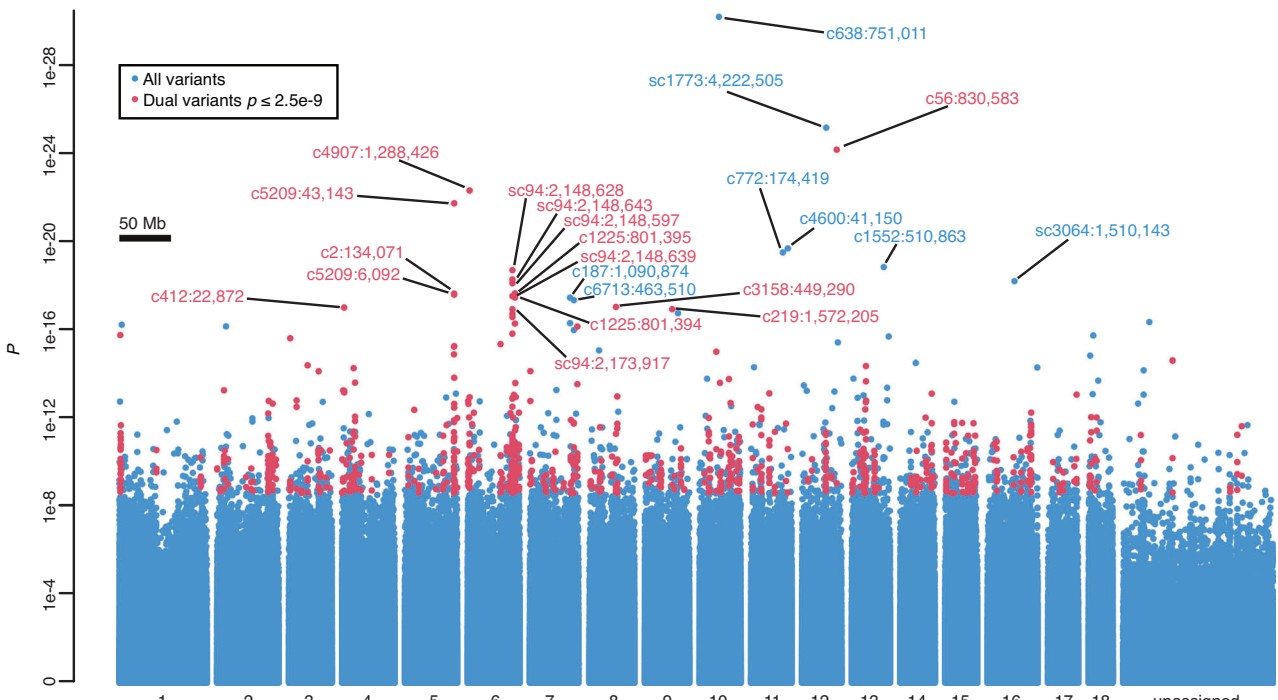

**Fig. 1 | Results of the pooled genome-wide association study (pooled-GWAS) identifying variants associated with resistance to *Schistosoma mansoni* in the *Biomphalaria sudanica* genome.** Fisher's exact test $p$ values for all pooled-GWAS variants are arranged horizontally based on contig orthology to 18 chromosomes (x-axis labels) of the *B. glabrata* genome (xgBioGlab47.1, NCBI RefSeq: GCF_947242115.1) and linkage map analysis (Dataset S2, Figs. S8, S11, S16). Pooled-GWAS dual-variants, defined as two or more proximate (<50 kb and >1.5 kb apart) variants strongly associated with *S. mansoni* resistance (uncorrected two-sided $p \le$ 2.5e-9) are red. All others are blue. All dual-variants and singleton-variants with uncorrected two-sided $p \le$ 1e-17 are labeled red and blue, respectively, with their contig and contig position. Unassigned contigs could not be unambiguously mapped and are mostly small and/or repetitive.

(<50 kb and >1.5 kb apart to ensure support from distinct read pairs) variants are strongly associated with *S. mansoni* resistance (arbitrary threshold of $p \le 2.5e\text{-}9$); such sliding dual-variant 50 kb windows encompass 18.625 Mb (2%) of the *B. sudanica* reference genome Bs111[27] and contain 888 (46%) of variants with $p \le 2.5e\text{-}9$.

## Amplicon panel genotyping reveals population structure and validates variants associated with resistance

A multiplex amplicon panel was designed using the Genotyping-in-Thousands by sequencing method[30] to genotype variants in individual snails at 234 dual-variants and 12 singleton-variants with $p < 1e\text{-}13$ identified from the pooled-GWAS analysis (Supplementary Data 2). The amplicon panel also contained 22 markers for a priori gene candidates and 201 'neutral' markers to facilitate a linkage map for improved *B. sudanica* genome assembly (Supplementary Data 3).

An independent set of 122 positive and 98 negative snails not included in the pooled-GWAS were reserved for validation of genomic variants via genotyping with the amplicon panel and are henceforth referred to as genotyped-validation snails. These independent genotyped-validation snails were used to validate the differential allele frequencies associated with *S. mansoni* resistance observed in the pooled-GWAS sequencing data. These were combined with a subset of the pooled-GWAS snails (genotyped-pooled-GWAS snails: 138 positive, 138 negative) for more precise estimates of ancestry and genotype frequencies.

The amplicon panel was successful (Supplementary Data 4), with few missing genotypes per individual (median 1.7%, range 0.2–100%; 90% interval 0.6–6%) or per locus (median 1.6%; range 0.2–78%; 90% interval 0.8–14.4%; Fig. S4). A signal of population structure was evident in both PCA (Fig. S5) and ADMIXTURE[31] analysis (Fig. 2A; Fig. S6). With K = 2 ancestral populations (CV = 0.60, versus 0.64 for K = 1), ancestry from Population 2 ranged continuously from 0 to 1 and was significantly correlated with resistance (logistic regression $p < 1e\text{-}10$). Mean Population 2 ancestry was 45% for positives and 66% for negatives. Among snails with <10% Population 2 ancestry ("Group A"), 73% were positive for infection. Among snails with >90% Population 2 ancestry ("Group B"), 26% were positive for infection. Thus, any marker differing in frequency between these ancestral populations could be correlated with infection phenotype, even if not linked to an etiological variant, and therefore could explain some outliers identified by the pooled-GWAS. A PCA of GWAS snails plus *B. choanomphala*, a closely related deep-water taxon/eco-phenotype of *B. sudanica* also collected in Lake Victoria[32], and *B. sudanica* inbred lines originating from Lake Victoria[27], shows that Population 2 is not similar to either of these reference taxa, and thus the GWAS ancestry signal is not caused by interspecies hybridization (Fig. S7).

After accounting for ancestry, only variants within two genomic regions, henceforth referred to as *SudRes1* and *SudRes2*, showed significance (Bonferroni-corrected $p < 0.05$, designated henceforth as 'validated-variants') in an ancestry-specific (Fig. 2B) or a regression (Fig. 2C) model (Supplementary Data 2). Notably, the p-values of the variants in *SudRes1* and *SudRes2* were amongst the lowest of the dual-variant outliers identified by the pooled-GWAS (Fig. 1). The top validated-variants for *SudRes1* (Fig. 2B) and *SudRes2* (Fig. 2C) acted as dominant markers (only two genotypes observed, possibly due to co-amplification of paralogs), so to assess genotype-phenotype associations in more depth we examined codominant proxy variants and used these as representative variants for each region (Fig. 2D). For the *SudRes1* representative variant (c582:65,596, Fig. 2D), allele T was protective in the pooled-GWAS and in the genotyped-validation snails. Combining data from the genotyped-validation and genotyped-pooled-GWAS snails, odds of *B. sudanica* infection with *S. mansoni* were 0.31 for genotype TT, 0.57 for genotype GT, and 1.37 for genotype GG. For the *SudRes2* representative variant (sc94:2,174,117, Fig. 2D), allele A was protective in the pooled-GWAS and in the genotyped-

validation snails. Combining data from the genotyped-validation and genotyped-pooled-GWAS snails, odds of *S. mansoni* infection in *B. sudanica* were 0.36 for genotype AA, 0.74 for genotype TA, and 2.40 for genotype TT. In our linkage cross, *SudRes1* was not significantly correlated with infection due to presence of a null allele in one parent but *SudRes2* was significantly correlated with infection (Fisher's exact test, $p < 0.01$; Figs. S8, S9) in the same direction as the GWAS and validation snails. An alternate linear mixed model analysis that accounts for close relatives in the validation set confirmed the signals of markers on *SudRes1* and *SudRes2* as well as a third locus on c219 which we do not pursue further here (Fig. S10).

The best-fitting multiple regression model (AIC = 463.25 and $p < 0.01$ for all variables) included: ancestry; *SudRes1* (representative variant c582:65,596, Fig. 2D); and *SudRes2* (representative variant sc94:2,174,117, Fig. 2D), with both genetic markers modeled as acting dominantly to avoid potential errors in inferring diploid genotypes caused by unseen deletions or duplications (Fig. 2E). The model predicts a > 2-fold effect of each genetic marker (2.5 and 3.5), and a 5.7-fold effect of ancestry. Thus, the predicted odds of infection for a snail with no Population 2 ancestry and major allele homozygous genotypes at *SudRes1* and *SudRes2* (3.86) is 49-fold higher (approximately $2.5 \times 3.5 \times 5.7$) than the odds for a snail with 100% Population 2 ancestry and minor allele homozygous genotypes at both loci (0.08).

## *SudRes1* is rich in paralogous genes encoding MEGF domains

*SudRes1* comprises 1.07 Mb of Bs111 and contains 23 protein coding genes across five contigs (c6844, c582, c5209, c6, c2) that are closely linked on chromosome 5 (Fig. 3A, Fig. S8 and Fig. S11). Notably, 10 of these 23 genes encode multiple epidermal growth factor (MEGF) domains (Fig. 3A, Supplementary Data 5). Three of these MEGF proteins in c6844, c5209, and c582, display a common single pass transmembrane domain (TMD) structure, with intracellular tyrosine-specific protein phosphatase (PTP) domains and extracellular MEGF and a galactose binding domain (GBD), forming a receptor-like PTP (RPTP) protein (Fig. 3B). Each of these three RPTP genes within the *SudRes1* region are adjacent to antistasin-like protein coding genes (Fig. 3C). Only 14 other MEGF/GBD-containing RPTP genes are annotated in Bs111, 13 of which are clustered near *SudRes1* on chromosome 5 (Fig. 3A, Supplementary Data 5).

The two validated-variants in *SudRes1* (Fig. 2B) were both contained within introns of MEGF/GBD-containing RPTP protein genes, in contigs c6844 (BSUD.17727) and c582 (BSUD.15164) (Fig. 3C, Fig. S12A, Supplementary Data 5). Similarly, of the three 'top-outlier' variants (defined as variants with $1e\text{-}04 < p \le 1e\text{-}03$ following validation, see Fig. 2B) in adjacent contigs c5209 and c6 in *SudRes1*, one occurred in intergenic sequence near a third MEGF/GBD-containing RPTP protein gene (BSUD.13443) and two were in noncoding sequencing in or near another gene expressing MEGF/GBD-containing protein (contig c6 ortholog 1, Supplementary Data 5).

Three of the five contigs in the *SudRes1* region, c6844, c5209 and c582, are homologous with each other and match the same unduplicated orthologous region on *B. glabrata* chromosome 5 and *B. pfeifferi* LG5 (Figs. S13 and S14). Aligned read coverage was also atypically low across all *SudRes1* contigs compared to the rest of the genome (Fig. S15).

We compared the Bs111 reference genome, harboring the susceptible *SudRes1* haplotype, to a PacBio genome assembled from a snail homozygous for a resistant *SudRes1* haplotype (Bs2280, coverage of ~13x, N50 of ~87 kb). While *SudRes1* is not fully assembled in either genome, we can detect substantial structural rearrangements resulting in different numbers of genes for some clusters of homologous loci (Supplementary Data 5, Fig. 3C). Consistent with extensive sequence duplication, Bs2280 includes multiple copies of some amplicon sites (Supplementary Data 6), explaining why these failed to show Mendelian segregation and instead acted as dominant

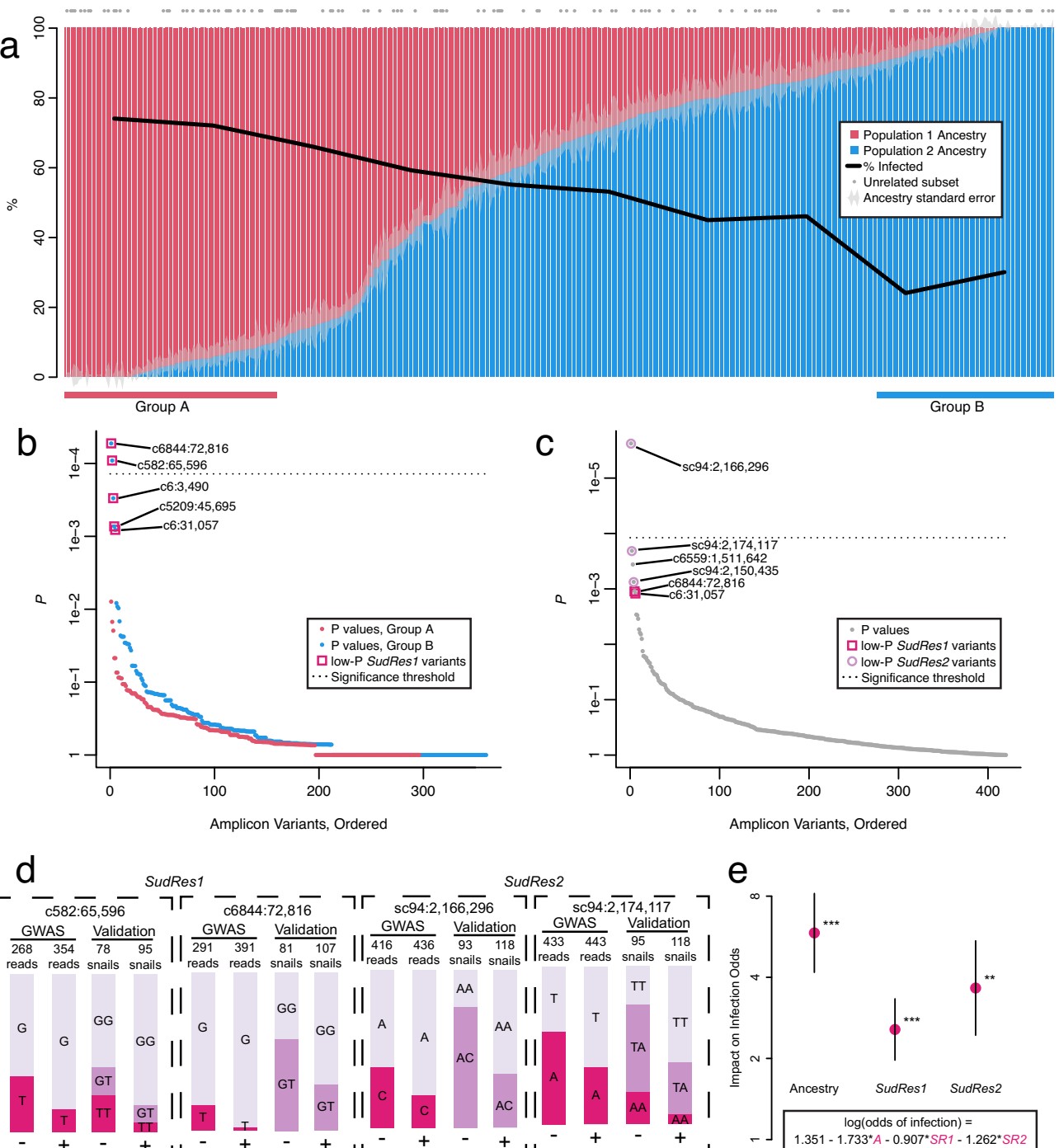

**Fig. 2 | Amplicon panel validation of pooled-GWAS results. a** Proportion of ancestry from *Biomphalaria sudanica* Population 1 (red) or Population 2 (blue) as predicted by ADMIXTURE[31] framed on a subset of unrelated individuals (represented by gray dots) is correlated with *Schistosoma mansoni* infection [Logistic regression, effect on odds of infection = 1.79 +/− 0.28, uncorrected two-sided $p = 8e-11$; $n = 496$ biological replicates (positive n = 260, negative $n = 236$)], such that >90% Population 1 snails (Group A) are 73% positive while >90% Population 2 snails (Group B) are 26% positive. Ancestry analysis sample size was $n = 503$ including 496 genotyped-validation and genotyped-pooled-GWAS *B. sudanica* and seven false negative *B. sudanica*. **b** Ordered Fisher's exact test *p* values per variant of genotyped-validation samples within ancestry groups. *SudRes1* variants c582:65,596 and c6844:72,816 are significant validated-variants (uncorrected two-sided $p = 5e-05$ and $9e-05$, respectively; Bonferroni adjusted significance threshold shown by dotted line) within ancestry Group B ($n = 34$ biological replicates, blue dots) and all other top-outliers (1e-04 $<p \leq$ 1e-03) are also in *SudRes1*. **c** Ordered regression *p* values per variant of genotyped-validation samples, after accounting

for ancestry ($n = 220$ biological replicates). *SudRes2* variant sc94:2,166,296 is a significant validated-variant (logistic regression, uncorrected two-sided $p = 2e-06$, Bonferroni adjusted significance threshold shown by dotted line) and most other top-outliers (1e-04$<p \leq$ 1e-03) are in *SudRes1* or *SudRes2*. **d** Allele and genotype counts for the most significant validated-variants from *SudRes1* (c6844:72,816) and *SudRes2* (sc94:2,166,296), which were both dominant markers, and the representative codominant marker variants from *SudRes1* (c582:65,596) and *SudRes2* (sc94:2,174,117), for *S. mansoni* infection positive (+) and negative (-) snails. Values are read counts for each allele in the pooled-GWAS ("GWAS"), and genotype counts in genotyped-validation snails ("Validation"). **e** Means and standard errors for each variable in the best-fitting multiple regression model (shown in box). Ancestry (A) is the proportion of Population 2 ancestry, and loci *SudRes1* (SR1) and *SudRes2* (SR2) are dominant effects at the representative codominant markers c582:65,596 and sc94:2,174,117. \*\*$p < 0.01$, \*\*\*$p < 0.001$. Logistic regression, uncorrected two-sided $p = 3e-07$, 5e-04, and 2e-03, respectively; $n = 380$ biological replicates (positive $n = 191$, negative $n = 189$).

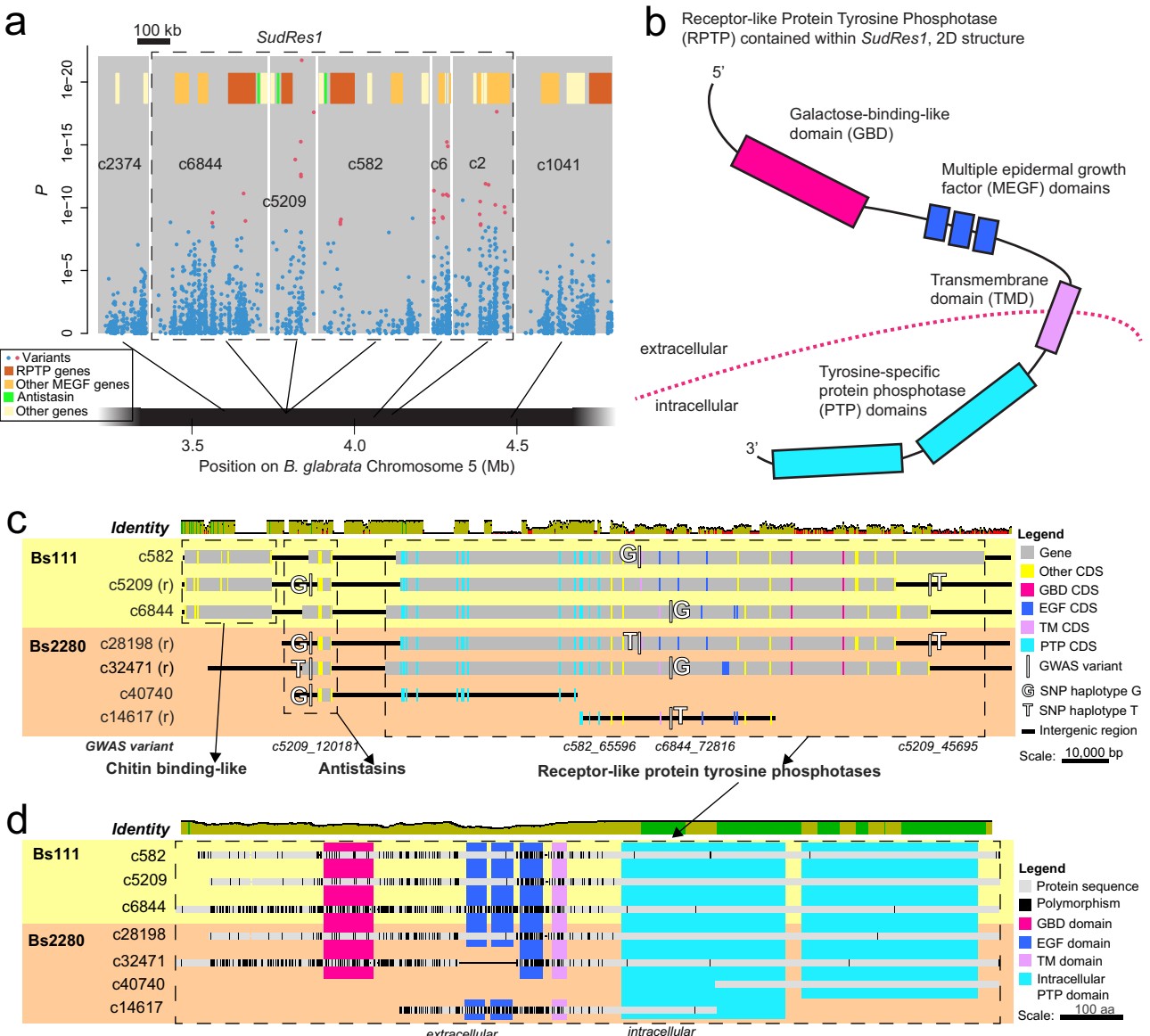

**Fig. 3 | Characterization of *Biomphalaria sudanica* SudRes1 genomic region.**
**a** Pooled-GWAS uncorrected two-sided *p* values (Fisher's exact test) in *SudRes1* regions (dashed boxes), which contain pooled-GWAS dual-variants (red) and other variants (blue), defined as in Fig. 1. Contigs (gray rectangles) on *B. sudanica* chromosome 5 arranged horizontally based on contig orthology to 18 chromosomes of the *B. glabrata* genome (xgBioGlab47.1, NCBI RefSeq: GCF_947242115.1) and linkage map analysis (Dataset S2, Figs. S8, S11, S16). Gene positions are shown (yellow/brown/green/orange boxes), highlighting particularly prevalent classes of genes: in *SudRes1* the multiple epidermal growth factor (MEGF) and galactose-binding like domain (GBD) containing receptor-like tyrosine-specific protein phosphatase (RPTP), other protein coding genes containing MEGF domains. **b** The predicted protein structure of receptor-like tyrosine-specific protein phosphatase (RPTP) coding gene BSUD.17727 (c6844) present in the *SudRes1* region of the *B. sudanica* genome and containing intronic validated-variants (Fig. S12A). A similar RPTP coding gene is contained within adjacent contigs c582 and c5209 within *SudRes1*, and contig c1041 neighboring *SudRes1* region (Fig. 3A). **c** Nucleotide alignment of paralogous contigs representing a portion of the *SudRes1* region in both Bs111 and Bs2280 genomes, showing location of genes and coding regions relative to GWAS variants used in the amplicon panel. Four G/T polymorphisms are shown, two of which act as non-Mendelian dominant markers due to paralogous amplification, one of which shows two alleles with Mendelian segregation (c582:65,596), and one of which segregates in our population but is invariant in these sequenced genomes (c5209:45,695). Exons are colored by protein domain, and mean pairwise sequence identity calculated in Geneious v2022.0.2 (Biomatters Ltd.) is shown in 100 bp sliding windows across aligned contigs (green = 100% identity, brown = 30% to <100%, red = <30%). **d** Amino acid alignment of RPTP genes in *SudRes1*, demonstrating extracellular diversity and loss of functional EGF domains in Bs2280 contig c32471. Protein domains are indicated by color, and non-synonymous polymorphisms in comparison to the majority consensus of paralogs/orthologs are shown in black. Sequence identity calculated in Geneious v2022.0.2 (Biomatters Ltd.) is shown as in (**c**), for 25 aa sligning windows in aligned proteins.

markers. One particularly variable segment, occurring in several divergent copies in both genomes, contains the adjacent antistasin and RPTP genes (Fig. 3C). Among putative orthologs, there are many nonsynonymous differences including occasional differences in protein length resulting in loss of functional domains, especially in RPTPs (Fig. 3D). Notably, this includes EGF domain loss in the resistant snail Bs2280 (Fig. 3D), suggesting that the snail's mechanism of *S. mansoni* resistance could involve the loss of function of a protein critical for parasite invasion.

## *SudRes2* is characterized by a large family of GRL101-like GPCR genes
*SudRes2* comprises a 440 kb region between 1.82 and 2.26 Mb on contig sc94 of Bs111 chromosome 6 (Fig. 4A, Figs. S8 and S16).

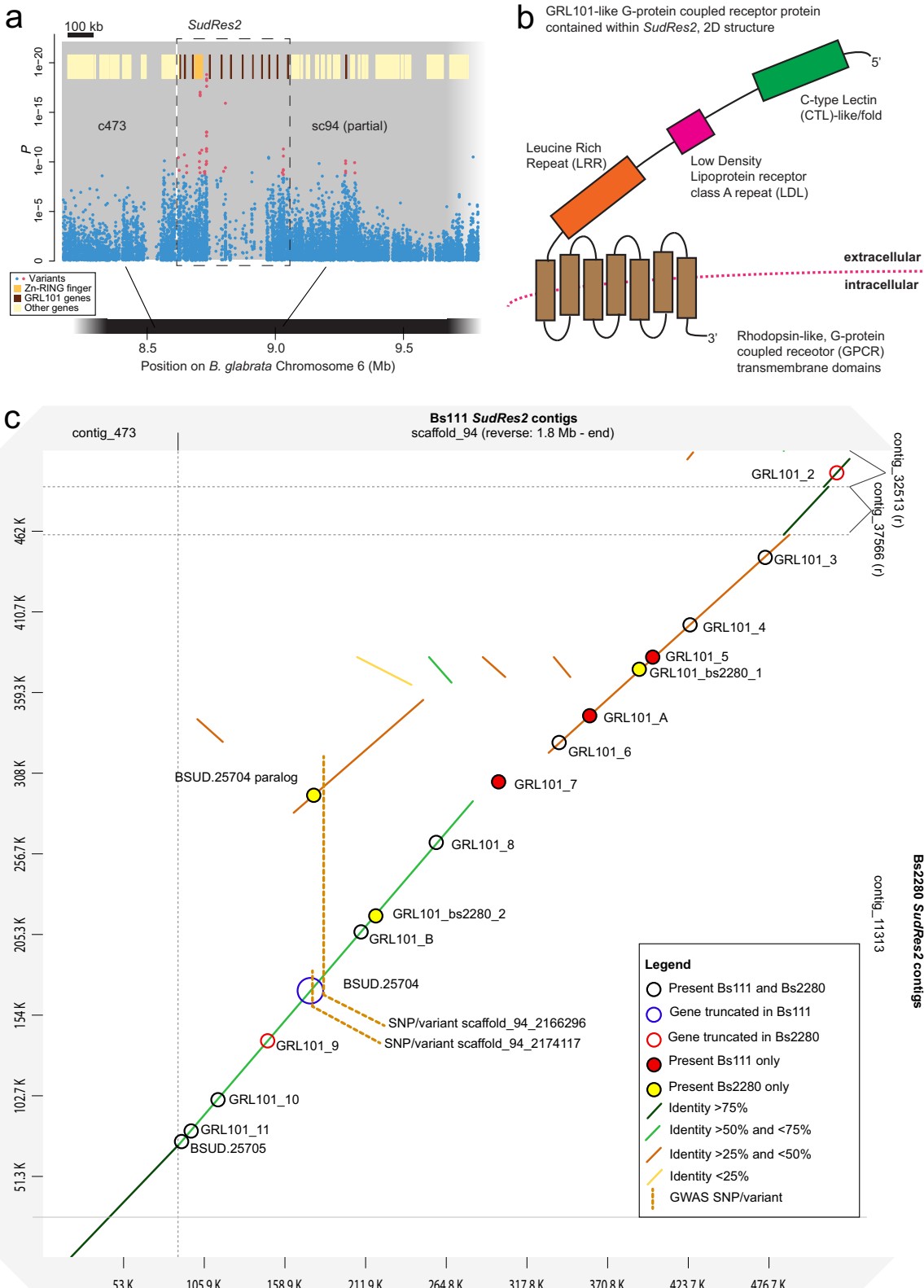

Following manual annotation of *SudRes2*, 14 protein coding genes were identified (Fig. S17 and Supplementary Data 7). Ten of these encode mutually paralogous GRL101-like proteins, defined as G-protein coupled receptor (GPCR) transmembrane proteins with extracellular regions containing a leucine rich repeat (LRR) region, a low-density lipoprotein receptor class A repeat (LDL) and a C-type lectin-like (CTL) domain (Fig. 4B); an additional two GRL101-like

genes in *SudRes2* are missing the CTL or LDL domain and may represent incomplete proteins (Supplementary Data 7). Of the 437 GPCR genes in Bs111[27], only six genes outside of *SudRes2* are annotated as possessing GRL101, CTL, and LDL domains (Supplementary Data 7).

Bs2280, the resistant snail genome, was also homozygous for a resistant *SudRes2* haplotype, facilitating comparison with the

**Fig. 4 | Characterization of *Biomphalaria sudanica* SudRes2 genomic region.**
**a** Pooled-GWAS uncorrected two-sided *p* values (Fisher's exact test) in *SudRes2* regions (dashed boxes), which contain pooled-GWAS dual-variants (red) and other variants (blue), defined as in Fig. 1. Contigs (gray rectangles) on *B. sudanica* chromosome 6 are arranged horizontally based on contig orthology to 18 chromosomes of the *B. glabrata* genome (xgBioGlab47.1, NCBI RefSeq: GCF_947242115.1) and linkage map analysis (Dataset S2, Figs. S8, S11, S16). Gene positions are shown (yellow/red/orange boxes), highlighting particularly prevalent classes of genes in *SudRes2* encoding a class of leucine-rich repeat containing G-protein couple receptors (GRL101) with C-type lectin and low-density lipoprotein extracellular domains (partial GRL101 genes included), and a Zinc-RING finger and inhibitor of apoptosis containing protein. **b** A representative predicted protein structure of a GRL101-like G-protein coupled receptor coding gene, twelve of which were predicted through manual annotation within the *SudRes2* region of contig sc94 (1.82–2.26 Mb) in the *B. sudanica* genome. **c** Dot plot constructed using D-GENIES[88] comparing synteny of the *SudRes2* region in *Biomphalaria sudanica* Bs111 genome[27] and the Bs2280 resistant snail genome, highlighting regions of divergence and structural rearrangements between the two genomes. GRL101 genes are indicated along with Zn-RING-IAP gene *BSUD.25704* and its paralog.

susceptible Bs111 haplotype. In both genomes, assembly of this region is nearly complete, and reveals several genes that are present in only one genome, or are duplicated in one haplotype (Fig. 4C). Similarly, some orthologs differed in length between genomes, being truncated in one or the other (Fig. 4C).

Both the validated-variant and highest top-outlier variant in Bs111 *SudRes2* are contained in the non-coding regions of non-GRL101 gene, *BSUD.25704*, clustered within the GRL101 genes, which when complete encodes a protein with a zinc finger RING-type (Zn-RING) domain and inhibitor of apoptosis (IAP) repeat region (i.e., Zn-RING-IAP) (Fig. 4C, Fig. S12B). In the reference/susceptible *SudRes2* haplotype, a nonsense variant (Bs111 sc94:2,167,458) in *BSUD.25704* truncates the protein at 323 aa, however in the resistant Bs2280 ortholog, a 391 aa protein can be translated. Furthermore, in Bs2280 a paralogous Zn-RING coding gene (truncated and not including IAP) is present within a divergent portion of the orthologous *SudRes2* region (Fig. 4C). Amplification of both *BSUD.25704* and its paralog in resistant haplotypes is likely responsible for the non-Mendelian behavior of the validated marker (sc94:2,166,296) which appears as heterozygous in resistant snails (Fig. 4C; Supplementary Data 6). On the boundary of *SudRes2* is a baculoviral IAP repeat containing (BIRC) protein coding gene *BSUD.25705* (Fig. 4C), many of which are contained in the genome regions neighboring *SudRes2*.

Prior to the manual annotation of the 14 genes contained with the Bs111 genome, the *SudRes2* region was exceptional in that: (1) only four protein coding genes in 440 kb had been annotated in this region of the reference *B. sudanica* genome, much lower than the genome-wide density of one gene per 40 kb; (2) a low density of variants were present (Fig. 3A and Fig. S17), and; (3) a large drop in aligned pooled-GWAS read coverage across the central 240 kb of *SudRes2* was apparent (Figs. S18 and S19). To confirm the presence and validity of the manually annotated GRL101 genes, *B. sudanica* RNA transcript data was successfully aligned to 11 of the 12 predicted GRL101 CDS sequences in the *SudRes2* region (all except GRL101_3). Phylogenetic analysis of protein coding sequences also revealed that the *B. sudanica* syntenic (conserved gene order) orthologs identified in *B. glabrata* and *B. pfeifferi* (Supplementary Data 7) were also the most closely related (Fig. S20).

## Discussion

In this study, we identified and validated two previously uncharacterized genomic regions, *SudRes1* and *SudRes2*, in the African snail vector *B. sudanica* that are associated with resistance to *S. mansoni* infection, each contributing a similar effect size over 2-fold to the *S. mansoni* infection odds ratio. Both regions contain long segments with unusually low pooled-GWAS read coverage and contain few annotated genes in the reference genomes of *Biomphalaria* sp., suggesting possible structural variation or allelic divergence that preclude unambiguous alignment of reads and complicates assembly and annotation of these regions. It is crucial therefore to acknowledge that the validated-variants associated with schistosome resistance may not themselves be causal polymorphisms, instead they highlight that something significant is occurring in these regions that may remain elusive using the current *B. sudanica* genome assemblies[27], potentially due to structure

rearrangements or unaligned alleles. Manual annotation of both *SudRes* regions revealed that they are enriched with transmembrane protein coding genes with diverse extracellular regions composing of protein-protein interacting and carbohydrate binding domains, relevant to immune-related functions such as pathogen recognition[33,34].

*SudRes1* is characterized by MEGF-domain containing genes, including receptor-like protein-tyrosine phosphatases (RPTPs), comprising extracellular MEGF, extracellular GBD, and intracellular tandem PTP domains. These potentially heavily glycosylated RPTPs may form stable dimers on the cell surface[35], with ligand binding triggering conformational changes that expose or occlude catalytically active regions of the intracellular membrane-proximal PTPs, transducing signals across the cell membrane[36,37]. The presence of schistosome resistance-associated variants surrounding the *B. sudanica* RPTPs suggests that increased efficacy or upregulation of these proteins may counteract *S. mansoni*-induced phosphorylation, one of the parasite's strategies to manipulate or evade the snail immune system and promote its survival[38]. Neighboring each RPTP in the *SudRes1* region were antistasin genes, a type of serine protease inhibitors that were originally described as anticoagulants in blood feeding species and since been attributed to immune responses in marine gastropods[39]. While *SudRes1* is not fully assembled in either Bs111 and Bs2280 genome, it appears likely that homologous contigs within each genome represent paralogous segments rather than alleles, since similar gene counts were observed in both genomes, suggesting recent gene duplications after *B. sudanica* diverged from *B. glabrata* and *B. pfeifferi*. Prior to its identification in this GWAS, the region now designated as *SudRes1* was previously noted as showing exceptionally high nucleotide diversity[27], which when coupled with the pooled-GWAS results suggest that pathogen-mediated balancing selection may act on these genes as previously hypothesized. Our findings here support the approach of using genome hyperdiversity as a proxy for identifying genes possibly related to immunity in uncharacterized genomes[27].

*SudRes2* contains resistance-associated variants within a Zn-RING-IAP gene, which neighbors a cluster of 12 leucine rich repeat-containing G protein-coupled receptor (LGR) family genes where pooled-GWAS variants are distributed throughout. The structure of the *SudRes2* LGR proteins is similar to GRL101, a LGR first described in the gastropod species *Lymnaea stagnalis* notable for its N-terminal extracellular LRRs and LDLs (UniProt accession P46023[40]). Unique to the *B. sudanica* GRL101 genes characterized in *SudRes2*, however, is the N-terminal C-type lectin (CTL) fold/domain. CTL domain containing proteins are established components of both vertebrate and invertebrate innate immune systems as recognition and effector molecules, which show pathogen dependent expression patterns[41–43]. Due to the architecture of the *SudRes2* GRL101 proteins, the CTL domain likely extends away from the cell membrane exposing the CTL binding region to cytoplasmic ligands (such as those derived from invading pathogens) that could then be presented to the GPCR membrane-spanning binding pocket, triggering G-protein activation. Homology and phylogenetic placement of the syntenic GRL101 proteins indicates a shared ancestry, and possible functional conservation, in GRL101 genes retained since the split of *B. glabrata* and African *Biomphalaria* species ~5 Mya[26,44], although the incomplete assembly of available

*Biomphalaria* genomes in this hyperdiverse region may impede inferences of expansion and contraction. To our knowledge, GRL101-like proteins have not been affiliated with immunity in gastropods, but have been shown to play an important role in innate immunity of other invertebrates[45,46]. Although GRL101-like genes were present elsewhere in the *B. sudanica* genome, the dense cluster of GRL101 genes in the *SudRes2* region is unique in that in Bs111 it is the only region <0.5 Mb with 12 GRL101 genes, with the caveat that GRL101 genes elsewhere in the *B. sudanica* reference genome may also not be annotated correctly.

A surprising result was the discovery of ancestry heterogeneity in our GWAS snails, whose parents had all been collected at the same time and place. More remarkable still, this ancestry signal is strongly correlated with schistosome resistance. Thus, many outliers in our pooled-GWAS could represent ancestry-informative markers with no physical linkage to resistance genes. Population ancestry estimates using only neutral linkage map markers were robust with low standard errors, supporting that the ancestry effect observed is real and not an artifact of atypical variants. Considering the importance of snail ancestry in schistosome compatibility here, potential causes behind the population structure were tested. First, no support for reproductively isolated cryptic *Biomphalaria* species in Lake Victoria causing the structure was found, since the ancestry estimates varied continuously between populations, and because no marker was fixed between ancestral populations. Second, the pattern is not due to the inclusion of close relatives, since ancestry estimates are framed upon an identified subset of unrelated individuals. Consistently, the estimated allele frequencies for the two ancestral populations are continuously distributed, and only few alleles are observed at a similar frequency, so we do not expect that a single prolific snail had parented a disproportionate amount of the offspring used in the GWAS. Third, this cryptic population structure does not align with divergence between *B. sudanica* and *B. choanomphala*, which have been considered sister species or ecomorphs of a single species in Lake Victoria[32,47]. Rather, Population 2 *B. sudanica* is distinguished by a set of alleles that do not appear to be common in either *B. sudanica or B. choanomphala*, and thus represents unique diversity that was not previously recognized.

The population structure of *B. sudanica* in Lake Victoria observed in our results is more consistent with historical isolation and reconnection of populations. Of the major lakes in the Albertine Rift Valley lake system (Victoria, Tanganyika, Malawi), Lake Victoria is a relatively young lake forming ~0.4 Mya, and has gone through at least three major desiccations in the past 100,000 years[48–50]. Shaped by such drought events, the cichlids of Lake Victoria have become a famous study system due to the astounding levels of explosive diversification that has occurred since the last desiccation event <15,000 years ago[51,52]. The population structure of *B. sudanica* observed suggests that indeed cryptic population structure is present, potentially caused by these historic events, yet the degree of admixing between populations in our study signifies ongoing outcrossing rather than clear speciation. The signatures of ongoing admixing may be influenced by hydrologic patterns of Lake Victoria. The collection site was ~20 miles north of the Rusinga channel connecting the open lake and narrow Winam gulf (Fig. S1). The Winam gulf is a unique lake environment given that it is somewhat separated from open lake water due to the prevailing currents limiting circulation of water[53,54], it is also comparatively shallower, potentially exacerbating historic water level changes, and has more protected shores, providing different freshwater habitats than those present in the open lake. The hydrology of the Rusinga channel and therefore Winam gulf was most recently disrupted by the blocking of the Mbita passage in the early 1980's, until its unblocking in 2017[55], therefore occurring just prior to our snail collections in early 2018. The return of north-easterly flow of open-lake water into the Winam gulf through the Mbita passage has caused a shift in both

bacterial and planktonic communities in the Winam gulf[56,57], and may have allowed dispersal of *B. sudanica* populations on floating vegetation, such as water hyacinth between lake areas[58]. Although we cannot establish the potentially different geographic origins of the *B. sudanica* representative of each population using currently available data, these snail population differences, and therefore vectoral competency differences, could explain why some locations around lakes are persistent hotspots of transmission while others are not[10]. These findings underscore how pathogen resistance can vary substantially between closely related populations, and in this instance could suggest that schistosome transmission may be more persistent in lake regions where highly susceptible *Biomphalaria* populations are present.

This study complements extensive work on immune mechanisms in laboratory populations of *B. glabrata*[17,20,22,59], facilitating comparisons between snail species. Notably, there was no overlap between our validated GWAS hits and loci linked to resistance in *B. glabrata*. The amplicon panel included at least two amplicons within or near each of these a priori candidates[27], and none of them showed a significant association with infection phenotype. *SudRes1* resides on chromosome 5, the site of a large resistance QTL in *B. glabrata*[22], though about 10 Mb away and thus unlikely to include the same gene(s). *SudRes2* resides on *B. sudanica* chromosome 6, which contains a density of schistosome-resistance a priori loci including *PTC1*[59], *tlr*[60], *sod1*[61], and the closest, *prx4*[62], at ~1 Mb away is not likely to be responsible for association in our analysis. While allelic variation in orthologs of *B. glabrata* resistance loci are not associated with *S. mansoni* resistance in *B. sudanica*, reverse genetics approaches successfully applied in *B. glabrata*[63,64] can be used in the future to functionally evaluate the roles of these genes. In addition, transcriptomic single-cell RNA-seq analyses of *B. glabrata* immune cells (hemocytes) have revealed potential immune effectors with hemocyte expression in this species, including 29 instances of MEGF-containing proteins[65,66], a gene family occurring repeatedly in *B. sudanica SudRes1*. *Biomphalaria sudanica* may rely on entirely different genetic mechanisms for parasite resistance than *B. glabrata*. However, considering the extensive genotype-by-genotype interaction documented between *B. glabrata* and *S. mansoni*[67], mediated by hyperdiverse resistance loci suggestive of long-term balancing selection[17,59], we propose a more nuanced scenario. Namely, that resistance alleles fluctuate dynamically in response to the genotypes of local parasites (including other trematodes), so the loci harboring common large-effect alleles will vary over time and space, even within a species. Other trematode species may be more prevalent and exert greater selection pressure on these snails, indirectly impacting resistance to *S. mansoni*. The striking 5.7-fold effect of ancestry on odds of infection also supports this dynamic view, as subpopulations different in resistance may be distributed unevenly across Lake Victoria.

Diversity at both *SudRes1* and *SudRes2* is high, as shown by previous polymorphism scans[27] and confirmed by the substantial sequence and structural divergence between susceptible genome Bs111 and resistant genome Bs2280. We are unable to pinpoint causal genes yet, and the numerous differences between resistant and suspectable haplotypes means several candidates are plausible. Each genome assembly contains genes and/or gene segments that are absent in the other assembly. Thus, the host-parasite interaction mechanism(s) could include recognition by *S. mansoni* of snail hosts possessing a particular susceptibility protein, triggering successful infection, or else recognition of the parasite by snails possessing a particular resistance protein, triggering immune cascades[68]. If what matters is parasite recognition of the host, loss of EGF domains in *SudRes1* RPTP proteins in resistant snails may inhibit parasite recognition of snail host molecules, and hinder parasite-driven modification of the host response. Similarly, loss of *SudRes2* GRL101 genes in resistant snails may prevent the parasite from recognizing these snails, rendering them immune. In contrast, if resistance is driven by host

recognition of the parasite, this could be mediated by nonsynonymous allelic differences in *SudRes1* and *SudRes2* or larger structural changes. For example, at *SudRes2* the non-truncated Zn-RING-IAP gene *BSUD.25704*, and its additional partial paralog, could fulfill a recognition function in resistant but not susceptible genomes. Parasite-resistance regions *PTC1*[59] and *PTC2*[17] in *B. glabrata* can show both dominant resistance and dominant susceptibility[59,69] and are also highly polymorphic in *B. sudanica*[27], indicating a similar pattern of immune-relevant balancing selection consistent with long-term snail-parasite coevolution.

By revealing immune-relevant genetic variation in *B. sudanica*, the primary vector in African Great Lakes, this work represents an important step toward molecular-informed vector control to combat schistosomiasis in high-transmission global regions, including gene drive technologies[70]. However, we demonstrate that genetic manipulation of snails for schistosomiasis control will require navigating the ever more complex genetic architecture of snail resistance, particularly when considering the non-overlap in findings from the laboratory model South American species *B. glabrata* and the diversity of snail vector species responsible for the majority of schistosome transmission in Sub-Saharan Africa. The impact of any resistance allele may vary due to genetic background and environmental factors. However, we expect that the large-effect loci identified here will play key roles in the continued elucidation of snail immunity as it pertains to human disease.

## Methods

### Ethical considerations
This project was undertaken following approval from the relevant bodies, including Kenya Medical Research Institute (KEMRI) Scientific Review Unit (Approval # KEMRI/RES/7/3/1 and KEMRI/SERU/CGHR/035/3864), Kenya's National Commission for Science, Technology, and Innovation (License # NACOSTI/P/15/9609/4270 and NACOSTI/P/22/14839), Kenya Wildlife Services (permit # 0004754 and # WRTI-0136–02-22), and National Environment, Management Authority (permit # NEMA/AGR/46/2014 – Registration # 0178 and NEMA/AGR/159/2022 – Registration # 201). Collections of *S. mansoni* from schoolchildren were approved by KEMRI's Scientific and Ethics Review Unit (SERU), reference SERU No. 3540, by Institutional Review Board of the Western University of Health Sciences No. FB19/IRB/111 2021, the and by the Institutional Review Board of the University of New Mexico (UNM) No.18115. Informed consent was obtained from the parents of five children that were deemed to be positive for *S. mansoni* from fecal samples using egg microscopy, of which remaining samples were used for *S. mansoni* miracidial hatching. All five children were treated with praziquantel during a follow-up visit by a physician.

### Snail phenotyping
In total, 329 *B. sudanica* snails were collected from Anyanga Beach Kanyibok, Kenya (Latitude: −00.08958°, Longitude: 34.08592°) on March 27th 2018. These snails were screened the following day for any patent echinostome infections, of which eight were found and discarded, leaving 321 snails that formed the parental generation. Parent snails were bred in plastic aquaria in an outdoor rearing facility at Kenya Medical Research Institute (KEMRI) in Kisumu. Snails housed in aquaria were allowed to breed and lay eggs for 7–14 days, then the adults moved to a new aquarium so that the eggs can hatch, and juveniles develop. This process was repeated every 7–14 days for approximately two months to reach the goal of 1400 offspring juvenile snails with a shell diameter of 4–6 mm to use in our infection cohort, which was reached in May 2018.

*Schistosoma mansoni* eggs were obtained from five local school children, aged 6–12 years old that attended the regional primary school, Kanyibok Primary School, Usenge, Kenya (Latitude: −00.08552°, Longitude: 34.07778°), on May 30th 2018, following informed consent from parents. Infected children were given praziquantel (40 mg/kg body weight) treatment during a follow-up visit by a KEMRI physician. Individual fecal samples were screened for *S. mansoni* eggs using the Kato-Katz method and then five samples with high egg intensity were selected for use in snail challenges that were performed over the course of the next two days, storing fecal samples at 4 °C overnight. The eggs hatch upon exposure to freshwater and light and the resulting free-swimming miracidia were used for snail exposures.

In total, 1400 offspring juvenile snails (4–6 mm) were each exposed to eight freshly-hatched miracidia in the wells of a 24-well tissue culture plates each filled with ~2 ml of bottled water overnight. Exposed snails were maintained in large plastic aquaria (~20 L) in an open-air snail facility under ambient light and fed lettuce. Each snail was checked for infection via cercarial shedding every three to five days from 5- to 9-weeks post exposure. Shedding snails were considered "true positive" and preserved in 100% ethanol. Any snails that did not release *S. mansoni* cercariae by 9-weeks post-exposure were later subjected to a PCR diagnostic to ensure they were "true negative" (uninfected) as opposed to harboring a prepatent/latent infection ("false negative")[29]. Only snails that were cercariae and PCR negative were considered negative and used for GWAS.

### Pooled genome-wide association study
Genomic DNA (gDNA) from each snail was extracted from headfoot tissue following a modified CTAB protocol for freshwater snails[71]. For the positive and negative pools, 10 ng of gDNA from each snail was included in the appropriate pool after quantification by two Qubit DS DNA assays (Invitrogen, CA, USA). Two Nextera libraries were created per pool as technical replicates to help ensure robustness and mitigate potential biases, and each was subjected to paired end 150 bp sequencing across two lanes of the Illumina NovaSeq 6000 S4 flow cell, with the positive and negative snail libraries on one lane and the technical replicate libraries of each on the other.

All command line and custom Perl scripts used in the following bioinformatic analysis are available in the FigShare project repository (https://figshare.com/projects/Genome_Wide_Association_Study_of_Biomphalaria_sudanica/230615). To process the resulting DNA sequence read data from each pool, Nextera Illumina Adapters were cut in *cutadapt* v3.1[72] and reads trimmed in *trimmomatic* v0.30 using options: LEADING:20 TRAILING:20 SLIDINGWINDOW:5:20 MINLEN:50[73]. *Schistosoma mansoni* reads were then removed from the DNA libraries by aligning data to the *S. mansoni* v9 genome (GCA_000237925.5)[74] using *bwa* v0.7.17 *mem*[75,76] and removing reads that mapped with CIGAR value > 70 using a custom Perl script (*FilterMappedBamCigar.pl*, options: -m 70) whilst removing secondary and supplementary reads in *samtools* v1.19.2 using options: -f 1 -F 2304 -bS[77] to retain only reads that are *B. sudanica*. FASTQ files of the filtered *B. sudanica* reads were generated using *BEDtools* function *bamtofastq*[78] then aligned to the *B. sudanica* genome[27] using *bwa mem*[75,76]. PCR duplicate reads were marked, and removed using *samtools fixmate* (options: -m) and *samtools markdup* (options: -s -f), along with unmapped, secondary and supplementary alignments, and only paired reads were kept (*samtools*[77] options: view -f 1 -F 3332 -bS). Due to large file size, some files had to be partitioned for improved processing, these were merged using *samtools merge*, before each bam file representing each library was indexed; *samtools index*.

The allele depth was used to calculate allele counts and therefore frequencies to analyze genotype-phenotype associations between our positive and negative pools of snails. First a Variant Call Format (VCF) file was generated using the *bcftools* v1.9[79] *mpileup* (options: -A -a AD) and *call* (options -m -Ov) functions. Subsequently, a custom Perl script that extracted information from the allele depth region (*AlleleCountsFromGwasVcf.pl*, options: -d 20 -q 0.05 -l 0,1 -s 100 -i), was used to make a table of counts of each allele for each sample in the VCF. The options correspond to filtering the variant data by

using a minimum allele count of 20, and removing variants if any of the four quality metrics (RPB, MQB, MQSB, and BQB) contained in the VCF were <0.05, or if allele depth was <100. Indels were removed. This resulted in a table of counts of each allele for each sample in the VCF. Technical replicates were compared by calculating the allele frequency difference between positive and negative pools at each variant, and regressing these values from one technical replicate against those from the other technical replicate. Technical replicates were then combined evenly in downstream analysis due to observed similarity.

For each variant in the *B. sudanica* genome that passed the filtering process, we calculated Fishers exact test $p$ values from $2 \times 2$ contingency tables representing allele count by phenotype (positive or negative), and the odds ratios for each variant, in R v4.3[80]. This resulted in a table denoting one row per variant with columns providing the allele count in positive and negative pools, odds ratio, Fisher's and Chi-squared $p$ values. This was then summarized using a custom Perl script (*SummarizeGWAS_p2.pl*).

*B. sudanica* contigs/scaffolds were ordered (Fig. 1) based on orthology to *B. glabrata* iM scaffolds and linkage groups[22], following our previous approach[27]. Namely, we used BLASTp (see **BLAST analysis**) of all *B. glabrata* proteins against all *B. sudanica* proteins, and vice versa, to identify reciprocal best hits. *B. sudanica* contigs/scaffolds for which a majority of ortholog-matched genes occurred on the same *B. glabrata* scaffold were assigned a position based on the location of those orthologs.

## Validation of variants by amplicon genotyping

To validate the pooled-GWAS outliers, a multiplex amplicon panel was designed using the Genotyping-in-Thousands by sequencing (GT-Seq) method[30]. The panel contains 470 amplicons in total, including markers for 234 dual-variants i.e., genomic locations with two or more proximate variants (<50 kb) that were strong outliers (Fishers exact test $p < 2.5e-9$), and 12 singleton-variants (with $p < 1e-13$ from the pooled-GWAS), and 22 markers for a priori gene candidates based on variants located near candidate immune genes of *B. glabrata* or exceptionally diverse regions of *B. sudanica* (panel details given in Supplementary Data 2)[27]. We applied the panel to the 220 independent validation snails (122 positive, 98 negative) and 276 of the pooled-GWAS snails (138 positive, 138 negative), referred to as genotyped-validation and genotyped-pooled-GWAS snails respectively, to confirm genotypes of snail individuals and validate the differential allele frequencies observed in the pooled-GWAS sequencing data.

In addition to the pooled-GWAS variants and a priori gene markers were an additional 201 'neutral' markers (i.e., with no expected phenotypic association) included in the amplicon panel to facilitate the inclusions of large contigs on a linkage map (see **Linkage mapping**, below). Also, a *Schistosoma* 16S rRNA (mitochondrial) amplicon to confirm parasite presence was included in the panel for future applications since a PCR diagnostic[29] was used to determine infected snails in this study.

In addition to the GWAS positive and negative *B. sudanica* genotyped by the amplicon panel, two outbred *B. choanomphala* snails collected in Lake Victoria using offshore deep-water dredging in 2019 from Ndiara Beach (Latitude:−00.090925°, Longitude:34.085067°) were also included for purposes of investigating population structure and utility of the developed panel on this taxon. We also genotyped seven "false negative" snails (did not shed parasites but PCR positive for infection) but we excluded these from phenotypic analysis due to their ambiguous status.

Amplicon genotyping was conducted by GTseek (Twin Falls, Idaho), which reports read counts of each allele per sample[30]. Sequencing libraries were prepared using the GT-seq genotyping method[30] modified by using Nate's Plates tagging and Normalization kits (https://gtseek.com/products/). Post sequencing, a custom Perl

script was used to count allele specific sequences for each locus from raw sequencing reads (https://github.com/GTseq/GTseek_utils). Amplicon genotypes were inferred from read counts as follows: <20x coverage considered missing, if each allele has >10% the coverage of the other allele the genotype is designated heterozygote, otherwise the genotype is designated homozygote (Fig. S4). We used Fisher's exact tests at each marker to confirm that genotype missingness was not systematically different between phenotype pools (fewer markers showed $p < 0.05$ than expected by chance, and their signal between pools was not in a consistent direction).

We assessed population structure in genotyped-validation snails and genotyped-pooled-GWAS snails using ADMIXTURE v1.3.0[31] and principal component analysis (PCA) in R. All command line, Perl and R scripts used in the following bioinformatic analysis are available in the FigShare project repository (https://figshare.com/projects/Genome_Wide_Association_Study_of_Biomphalaria_sudanica/230615). Specifically, we used the snpgdsIBDKING function from the SNPRelate v1.40.0 package[81] and PC-AiR in GENESIS v2.36.0[82] to identify a set of unrelated individuals, due to the possibility of close relatives in our set of F1 snails. We then ran ADMIXTURE on this unrelated set and projected the remaining samples onto the resulting ancestry inferences with the -P option. Standard errors were inferred by bootstrapping with the -B option. For both PCA and ADMIXTURE, we only used the 'neutral' linkage map markers rather than the GWAS outliers or a priori candidate markers. For ADMIXTURE we ran models with K (number of ancestral populations) ranging from 2 to 7 and chose the value of K with the lowest Cross-validation (CV) error in the unrelated samples. We also performed PCA including genotypes from the two outbred *B. choanomphala* samples, and the genotypes of two previously sequenced *B. sudanica* inbred line snail genomes[27]. We did not use all four Illumina-sequenced inbred lines as two were parents in our linkage cross and our markers were chosen to be different between them, causing a strong ascertainment bias. Given a signal of population structure, we accounted for ancestry in our validation analysis using two complementary statistical approaches. For both approaches, we converted genotype markers to two sets of binary variables, alternately designating heterozygotes as 0 or 1, effectively modeling a dominant effect on phenotype. We chose this approach rather than an additive model because deletions and duplications can lead to incorrect inferences about diploid genotypes[83], and we avoid this complication by simply asking if an allele is present in at least one copy, or absent. In our first approach, we fit a logistic regression model for each marker in the set of genotyped-validation samples, assuming a dominant effect on phenotype, and including ancestry as a quantitative cofactor (proportion of ancestry from second population). This was calculated in R using the glm function with family = binomial, with phenotype (binary 0 or 1) as the response variable, and ancestry (continuous from 0 to 1) and genotype (binary 0 or 1) as independent variables. In this way a genetic marker will only be significant if it has an effect on the phenotype that isn't already captured by the ancestry variable in the regression. We designated significance using a Bonferroni threshold of 0.05/2 N, where N is the number of candidate amplicons (i.e., excluding linkage map amplicons) with minor allele frequency >5% and <20% missing genotypes (≥176 samples with non-missing genotypes) in genotyped-validation samples, doubled to account for two tests per marker as either allele could be dominant. Second, to test for effects specific to ancestry groups, we divided genotyped-validation samples into two groups based on ancestry (group A and B), and performed Fisher's exact tests (in R with the fisher.test function) between genotypes and phenotypes within each ancestry group. To minimize ancestry effects, we required snails in these ancestry groups to show at least 90% ancestry from one ancestral population, and therefore ignored the more admixed samples. We designated a significance threshold as before, again requiring <20% missing genotypes per group (≥37 samples with non-missing genotypes in Group A;

≥28 samples with non-missing genotypes in Group B). The sample size used for Group A and Group B in this analysis is 46 and 34, respectively. To assess whether close family relationships drive these results, we first used snpgdsIBDKING[81] to identify snail pairs with estimated KING-robust kinship coefficients. For each marker we then fit a linear mixed model against the infection phenotype using the lmm.diago function from the gaston v1.6 package[84], incorporating the eigen decomposition of the kinship matrix with the eigenK argument. As above, we ran two different analyses: one including our previously-defined ancestry estimate as a covariable, and one restricted to just ancestry group A or B. We calculated $p$ values from the z-score obtained as BLUP_beta/sqrt(varbeta).

To find the best-fitting model relating genotype to phenotype from the amplicon panel data, we combined both genotyped-validation and genotyped-pooled-GWAS snail data. We tested all combinations of logistic multiple regression models (base $e$) encompassing ancestry and all successfully validated loci, considering all combinations of dominance at the genetic markers. We chose the model with the lowest AIC, and for which all individual variables have $p < 0.05$.

### BLAST analysis

BLAST searches (BLASTn, tBLASTn, BLASTp and BLASTx) described in the following sections were all performed using BLAST v2.14.0+ or BLAST v2.15.0+ using default parameters (E-value threshold 10 and scoring matrix of BLOSUM62). The queries consisted of nucleotide or protein sequences in FASTA format, and searches were run against the respective genome or protein databases created using the function 'makeblastdb'.

### Linkage mapping

To generate crossed snails for linkage mapping, two juvenile (<4 mm) *B. sudanica*, one from each line 163 and KEMRI, which demonstrate opposite patterns of susceptibility to *S. mansoni* (UNMKenya line)[85], were reared in a ~1 liter breeding tank. Once egg laying and hatched juveniles were present in the tank, the two parent snails and eight randomly selected juveniles had gDNA extracted using the Qiagen Blood & Tissue kit (Qiagen, MD, USA). Outcrossing was confirmed using a developed PCR and restriction enzyme based diagnostic marker[86]. Remaining F1 juveniles were moved to a new tank and then reared until eggs and F2 juveniles were present. These F2 juveniles were reared until >3 mm size. F2s were challenged with *S. mansoni* miracidia and classified as positive or negative for infection using the same methods as for the GWAS snails. gDNA of the Bs163 and BsKEMRI parents and 87 F2 *B. sudanica* snails were genotyped using the amplicon panel as described above.

Genotype data from parents and F2s were used to create genomic linkage maps with OneMap v3.0.0[87], using minimum logarithm of the odds (LOD) of 4.5 from the rf_2pts option to unite markers into linkage groups. We included all amplicon markers that segregated in the F2s, whether they were included as GWAS variants or linkage map markers. Amplicon sequences were then matched to the *B. glabrata* genome assembly (xgBioGlab47.1, NCBI RefSeq GCF_947242115.1) using BLASTn, facilitating synteny comparison between our *B. sudanica* map and the *B. glabrata* genome. We did not perform a map-wide analysis for quantitative trait loci due to low statistical power given our samples size. However, for the validated genomic regions from the GWAS, we tested for phenotype-genotype associations in the linkage cross using Fisher's exact tests.

### Characterization of validated genomic regions

Regions of the *B. sudanica* genome that were significantly associated with *S. mansoni* resistance and then validated were characterized using the *B. sudanica* genome annotation[27] to determine the function of protein coding genes, and annotate any potentially missing genes from the current *B. sudanica* annotation. Synteny of regions to the *B.*

*glabrata* genome assembly (xgBioGlab47.1, NCBI RefSeq GCF_947242115.1) and *B. pfeifferi* genome assembly UNM_Bpfe_1.0 (GenBank GCA_030265305.1) were assessed using D-GENIES[88]. tBLASTn searches against the *B. sudanica* genome were conducted using proteins in the syntenic regions of the *B. glabrata* genome. Alignments of *B. glabrata* and *B. pfeifferi* homologs/orthologs (see Dataset S5 and S7) to *B. sudanica* sequences were used where possible to determine complete gene sequences in the case that the reference protein sequences were suspected to be truncated. Open reading frames (ORFs) were assessed in the *B. sudanica* validated genomic regions using the in-built tool in Geneious v2022.0.2 (Biomatters Ltd.) and predicted protein sequences generated that were characterized based on the protein families and functional domains predicted using InterProScan[89] and DeepTMHMM[90]. Neighboring ORFs in the same direction were merged (internal stop codons in ORFs, predicted to be in intronic regions, were removed) to construct complete coding sequences of suspected proteins. Available PacBio and Illumina RNA-seq transcript data for *B. sudanica* (NCBI accessions: SRX22544968-SRX22544970) were aligned using *bwa mem*[75] to predicted mRNA sequences of these proteins.

BLASTn searches of predicted mRNA sequences generated following manual annotation were performed against the *B. sudanica* genome to determine any other homologous genes in the genome that may have not been annotated. For the 23,598 genes assigned to have open reading frames in the *B. sudanica* genome[27], these were searched against using key descriptors of genes of interest (i.e., PTPRA and GR101). In addition, InterPro accessions attributed to each gene in the annotated *B. sudanica* genome were searched against InterPro gene families and functional domains of interest. Specifically, to determine homologous genes to GRL101-like G-protein coupled receptor (GRL101) proteins containing a leucine rich repeat (LRR) region, a low-density lipoprotein receptor class A repeat (LDL) and a C-type lectin-like (CTL) domain, we filtered gene data by those matching InterPro accessions: IPR000276; IPR017452; IPR001304; IPR016187; IPR016186; IPR032675; IPR001611; IPR036055; IPR002172. To determine homologous genes to receptor-like protein-tyrosine-specific phosphatases (RPTP) containing multiple epidermal growth factor (MEGF) and galactose binding domain(s) (GBD), we filtered gene data by those matching InterPro accessions: IPR000242; IPR02902; IPR016130; IPR000387; IPR003595; IPR008979; IPR009030; IPR000742; IPR002049.

We examined read depth in our original pooled-GWAS data at our validated regions, to test if it was unusual relative to the rest of the genome. We calculated mean depth for each phenotype pool in 10 kb windows and normalized to the genome-wide median for that pool (Figs. S15, S18). We also calculated mean depth in 100 kb sliding windows (step size = 10 kb) and plotted this for positive and negative pools (Fig. S19).

### PacBio whole genome sequencing and assembly of a resistant *Biomphalaria sudanica* snail genome

High molecular weight DNA from a single snail included in the pooled-GWAS (Bs2280) that was phenotypically resistant (not shedding cercariae or *S. mansoni* PCR positive) and contained non-reference (resistant) haplotypes assessed using the amplicon panel at both *SudRes1* (contigs c582_65596: TT, contig_6_3490: AT and contig_6844_72816: GT, the latter two not indicating heterozygosity but rather paralogs, since there are never any TT's) and *SudRes2* loci (contig sc94 positions 2174117: AA, 2150435: CC and 2149367: AA) was sequenced on the PacBio platform. Genomic DNA was cleaned, fragmented and library produced following the standard HiFi SMRTbell Library preparation by Novogene (Sacramento, CA). The library was sequenced on a single PacBio Revio using a single SMRT cell, resulting in 13.4 G total bases. The PacBio ccs reads (raw reads available in NCBI BioProject: PRJNA1149315, BioSample: SAMN45084274) were

assembled using Flye[91] v2.9.4, settings flye --pacbio-hifi -g 1 g -t 16. Basic statistics of the genome assembly were calculated in seqkits v0.16.1[92].

Following assembly, orthologous contigs to *SudRes1* and *SudRes2* regions were identified by performing mutual best hit BLASTn searches for orthologous contigs to those in the Bs111 *SudRes1* and *Sudres2* region, in addition to tBLASTn and BLASTx searches to orthologous protein sequences. Orthologous regions were then aligned using MAFFT[93], and orthologous protein coding regions annotated in Geneious v2022.0.2 (Biomatters Ltd.).

### Reporting summary

Further information on research design is available in the Nature Portfolio Reporting Summary linked to this article.

## Data availability

The sequence data generated in this study have been deposited in the NCBI SRA database under BioProject accession code PRJNA1149315, including PacBio HiFi raw reads from genome sequence data of *B. sudanica* Bs2280 (SRS23462923), Illumina raw reads of infected (SRS22385825 and SRS22385826) and uninfected (SRS22385827 and SRS22385828) pooled-GWAS sequence data and amplicon panel Illumina raw reads of 503 individually genotyped *B. sudanica* (SRR32947997-SRR32948588 under SRP527155) and two *B. choanomphala* (SRR32948589 and SRR32948590). The genome assembly of Bs2280, generated for this study, is available on FigShare Project repository 230615. The genome assembly of Bs111 used in this study is available in the NCBI Genome database under BioProject accession code PRJNA1041389.

## Code availability

Perl and R scripts used for the analysis in this study, are available on FigShare Project repository 230615.

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

## Acknowledgements

This work was funded by National Institute of Health, National Institute of Allergy and Infectious Disease grants R01AI141862 and R37AI101438.

## Author contributions

J.A.T., E.S.L., M.R.O., and M.L.S. designed research; T.P., J.M.S., T.M., G.O., F.R., K.A., B.M., M.E.O., M.W.M., G.M.M., and M.L.S. performed research; T.P. and J.A.T. analyzed data; T.P., J.A.T., and M.L.S. wrote the paper.

## Competing interests

The authors declare no competing interests
