## [Transparent Peer Review file · Nature Communications]

Genes linked to schistosome resistance identified in a genome-wide association study of African snail vectors

Corresponding Author: Dr Michelle Steinauer

Version 0:

Reviewer comments:

Reviewer #1

(Remarks to the Author)

The manuscript addresses the urgent need for innovative strategies to control schistosomiasis, particularly by manipulating snail vector immunity. Through a genome-wide association study (GWAS) of African *Biomphalaria sudanica* populations, the authors identified two genomic loci, SudRes1 and SudRes2, linked to resistance against *Schistosoma mansoni*. These loci encode receptor-like protein tyrosine phosphatases and leucine-rich repeat-containing G-protein-coupled receptors, highlighting immune functions relevant to host-pathogen interactions. While this study represents a significant advancement in the field, several concerns must be addressed to ensure that its conclusions are well supported. If confirmed through more rigorous analysis, these findings could serve as a foundation for identifying genetic targets for vector control and advancing research on vector-based schistosomiasis interventions.

Major concerns:

1. To perform experimental *S. mansoni* infections (phenotyping) and subsequent genotyping experiments, 1,400 F1 snails derived from 329 parental snails were used. Even if all F1s were generated through cross-fertilization of unrelated parental snails, this experimental design would still result in the sampling of closely related siblings. However, neither the population structure analysis nor the genotype-phenotype association analysis adequately accounted for this familial relatedness among samples. Both PCA and ADMIXTURE analyses assume that samples are unrelated, and these methods for inferring genetic ancestry can fail when applied to related samples (PMID: 25810074). We strongly encourage the authors to use methods specifically developed for analyzing samples with familial relatedness, such as PC-AiR (<https://bioconductor.org/packages/devel/bioc/vignettes/GENESIS/inst/doc/pcair.html>). After identifying the subset of unrelated individuals in the dataset, PCA and ADMIXTURE analyses can be run using them as a reference set, while the remaining samples (which may be related to the reference set) can then be projected onto the resulting population structure.
2. PCA (Fig. S5) shows that PC2 (10% of the variance), rather than PC1 (42% of the variance), represents the ancestry component identified by ADMIXTURE (Fig. 2A). This indicates the presence of an orthogonal genetic principal component (PC1) with a higher variance contribution that cannot be attributed to the ADMIXTURE ancestry used in the regression modeling. This discrepancy raises concerns about the robustness of ancestry inference and must be addressed to ensure the validity of the downstream association analysis. It could be useful to inspect the PC loadings to ensure that they broadly represent variation across the genome, rather than one or a small number of genomic regions. Additionally, including a PCA plot with phenotype labels (i.e., resistant or susceptible), alongside ancestry proportions, could improve data interpretation.
3. Amplicon-seq data quality control (QC) needs to be more transparent and rigorous. Inclusion/exclusion thresholds for missingness per marker and per individual were overly lenient and inconsistently applied, varying substantially between analyses without clear justification. These thresholds as either minimum or maximum proportions should be reported, as appropriate, to improve transparency in both the main text and the supplementary material (e.g., "<10% missing" instead of the number of genotypes/individuals). Including additional QC plots for the amplicon data, similar to Fig. 4 in PMID: 25476721, would help readers assess data quality.
4. Two resistance-associated regions (SudRes1 and SudRes2) were identified, and both regions exhibit complex structural variations (Figs. 3 and 4). When SNPs overlap with CNVs (both deletions and duplications), the accuracy of genotype and allele counts may be compromised, potentially leading to spurious genotype-phenotype associations. Liu et al. (PMID:

30019117) clearly demonstrate in detail how the coexistence of CNVs and SNPs in the same genomic region can distort significance calculations in association studies. Importantly, this issue is not limited to pool-GWAS analysis; amplicon analysis of individual snails is also affected. For example, heterozygous deletions lead to a hemizygous allele at corresponding genomic regions, and these can be taken as homozygous, distorting their allele counts. The significance of SudRes1 and SudRes2 was largely supported by four SNPs, all of which overlap with CNVs. The authors generated PacBio and Illumina whole-genome sequencing data from individual snails and upon examination of their raw read alignments for evidence of CNV colocalization with significant SNPs, the following IGV screenshots were generated that display the samples in this order: Bsud111, Bs2280, BsKEMRI, Bs163, Bs110, and Bs5-2.

(1) c6:3,490 (most significant SudRes1 variant) colocalizes with a deletion CNV (Bs163).

(2) c582:65,596 (representative SudRes1 variant) colocalizes with a deletion CNV (Bs2280 and Bs163)

(3) sc94:2,166,296 (most significant SudRes2 variant) colocalizes with a duplication CNV (Fig. 4C). This SNP appears as a heterozygous variant in Bs2280 with ~2x coverage because reads originating from two paralogous regions are aligned to the same locus.

(4) sc94:2,174,117 (representative SudRes2 variant) colocalizes with a duplication CNV (Fig. 4C). Although this SNP appears as a homozygous variant in Bs2280, reads originating from two paralogous regions are aligned to the same locus (2x coverage). This duplication can be seen clearly in Fig. 4C. Sequence variations between paralogous copies may confound genotyping in this region.

The authors need to demonstrate that the association signals in SudRes1 and SudRes2 are supported by SNPs (in high LD with these regions but) located outside of CNVs and that they are not artifacts caused by distortions in allele/genotype counts due to the coexistence of CNVs and SNPs in the same genomic region. The authors could first identify CNV regions in the genome using WGS data and assembly comparisons and exclude these regions from SNP-based association analysis.

Minor concerns:

L100: Correlation between technical replicate pairs were assessed using Fisher's exact test p values (Fig.S1). This analysis ignores the direction of allele frequency differences. Correlating allele frequencies between replicates would be more informative.

L103: "In the pooled-GWAS results, genotype-phenotype association p values ranged as low as $1e-30$, including 45 variants (0.001%) with $p \leq 1e-15$ and 1,930 variants (0.04%) with $p \leq 2.5e-9$. (Fig. 1)." When Fisher's exact test is used in pool-seq analysis, read counts are used in place of true allele counts, which cannot be directly measured. Because 493 positive and 295 negative snails were pooled, the total ploidy in each pool was 986 and 590, respectively. When sequencing depth exceeds these ploidy counts, pool-seq p values may be lower than the true significance of allele counts differences. In such cases, p values cannot be simply interpreted as indicators of association strength, as they become influenced by sequencing depth (i.e., higher depth results in lower p values even when allele counts remain unchanged). This dependency of Fisher's exact test p values on sequencing depth also confounds the comparison of association signals between loci with unequal read depths in GWAS. Genomic read coverage fluctuates along chromosomes due to random distribution and biases such as GC content where high GC regions exhibit elevated coverage. Fisher's exact test is more likely to detect allele frequency differences in high coverage regions. One way to mitigate this bias is to subsample to a uniform coverage by randomly sampling bases before estimating the significance of allele frequency differences (PMID: 22025480).

L115: Methods section show describe how these 201 'neutral' markers were selected. Using the amplicon panel (which also contains *S. mansoni* targets), a linkage map was constructed by crossing parental snails that differ in their susceptibility to *S. mansoni* infection. Was this amplicon dataset used for pedigree-based mapping of resistance loci?

L124: "The amplicon panel data (of which median missing data was 2% per locus and 2% per individual) revealed a signal of population structure.." Instead of reporting the median missing rate, reporting of the minimum/maximum missing rates could be done for transparency. Were the inclusion/exclusion thresholds sufficient to ensure a robust analysis? A case/control nonrandom missingness test is often performed in GWAS. Fisher's exact test can be applied to missing call counts at each variant to check for systematic biases in the dataset.

L126: "With $K=2$ ancestral populations ($CV = 0.53$, versus 0.61 for $K=1$).." What were the CV values for $K>2$? This information could be included in a supplementary figure. In addition, ADMIXTURE can estimate parameter standard errors using bootstrapping and they can be displayed in Fig. 2A.

L135: This approach is likely not as robust as PCA or ADMIXTURE. Once genetic structure is modelled in ADMIXTURE using unrelated individuals, *B. choanomphala* and *B. sudanica* inbred lines can be "projected" onto the population structure to estimate their ancestry. The supplementary information mentions that a PCA analysis was performed including these samples, but the result is not included in the manuscript.

L141: It is questionable whether the analysis presented in Fig. 2B adequately accounted for ancestry, as ancestry

proportions vary continuously (Fig. 2A). Individual PCA plots for Group A and Group B with phenotype labels could help determine whether this approach effectively accounted for ancestry.

L147: “acted as dominant markers (only two genotypes observed)”. It is highly likely that only two genotypes were observed not because these variants acted as dominant markers but due to a genotyping error in which paralogous alleles were amplified by primers, resulting in the variants being incorrectly genotyped as heterozygotes. This must be clearly conveyed to the readers in the main text.

L428: Datasets S1–S6 were not accessible to reviewers and need to be made available. The genome assembly of Bs2280 is also not available on Figshare.

Reviewer #2

(Remarks to the Author)

Reviewer #3

(Remarks to the Author)

The manuscript describes a genome-wide association study of African *Biomphalaria* snails that are intermediate hosts of the parasite that causes schistosomiasis in an endemic area of high transmission. The study identified two genomic regions that were significantly associated with snail immunity. The significant aspect of this work is that it is the first time an exhausted genome-wide study has been done in a wild population of *Biomphalaria sudanica* snails with substantial statistical power since 493 infected and 295 uninfected snails were sequenced for this work. Furthermore, the study employs the most recent discoveries and advancements in genomics of this snail genus enhancing the understanding of host-parasite interactions in natural populations.

I read carefully the material and methods information provided with the manuscript, and I did not find any flaws in the study. Moreover, I found the approach very original, since it is very hard to obtain high-quality sequencing coverage per individual snail, therefore the GWAS-pooled approach is an efficient method to assess single nucleotide polymorphisms in a wild population. Furthermore, the authors also employed another genotyping technique that allowed them to validate their findings in the pooled sequencing approach which provides robustness to their data.

Besides the authors include in the discussion the limitations of their results highlighting that the two uncharacterized genomic regions associated with resistance to *S. mansoni* infection have low read coverage and few annotated genes.

I agree with the authors that this work represents an important step toward molecular-informed vector control to combat schistosomiasis in high-transmission regions and highlights the importance of the genetic background and environmental factors on the impact of resistance alleles.

While the data and overall methodology presented in the study are clear, I found the methods related to ancestry somewhat difficult to follow. Since not all readers may be familiar with population structure analysis, it would be helpful if the authors provided a more detailed explanation in the methodology section, specifically regarding how snail ancestry and schistosome resistance were evaluated. Specifically, in the paragraph where they mention: “We assessed population structure in genotyped-validation snails and genotyped-pooled-GWAS snails using ADMIXTURE (15) and principal component analysis (PCA) in R”, additional details are needed.

While Figure S5 displays three different subpopulations of snails, it is unclear why the samples are colored based on population 2 ancestry. It would be helpful if the authors could explain the significance of Population 2 in this context. Is this population used as a reference population?

Moreover, can the authors clarify how they validated that the variants identified in the GWAS analysis are specifically associated with snail immunity, rather than reflecting genetic differences between distinct population groups?

Providing this information would improve the transparency and reproducibility of the study.

The statistical methods and analytical approach in this manuscript are robust, and I do not have any significant suggestions for improvement. I also believe that no additional experiments or data are necessary for the study.

There are only a few minor adjustments to consider:

Figure 1 has an issue with the y-axis values that should be corrected.

Similarly, Figure 3A has the same issue with the y-axis values and requires adjustment.

The results are presented with substantial context and thoughtful integration of previous studies. However, the discussion could be further strengthened by including literature on cryptic population structures observed in other *Biomphalaria* species. This would provide additional insights and help hypothesize the causes of the cryptic population structure identified in this study. Some suggestions include:

Standley, C. J., Goodacre, S. L., Wade, C. M., & Stothard, J. R. (2014). The population genetic structure of *Biomphalaria choanophala* in Lake Victoria, East Africa: implications for schistosomiasis transmission. *Parasites & vectors*, 7, 1-10.

Sire, C., Langed, J., Barral, V., & Theron, A. (2001). Parasite (*Schistosoma mansoni*) and host (*Biomphalaria glabrata*)

genetic diversity: population structure in a fragmented landscape. *Parasitology*, 122(5), 545-554.

The manuscript would also benefit from the inclusion of additional references on immune mechanisms in laboratory populations of *Biomphalaria glabrata* mentioned in line 355. Specifically, citing studies that explore genomic regions associated with immune responses to schistosome parasites, as well as research on *Biomphalaria-Schistosoma* molecular interactions, would provide further support and enhance the contextualization of the findings.

A few suggestions are:

Mitta, G., Galinier, R., Tisseyre, P., Allienne, J. F., Girerd-Chambaz, Y., Guillou, F., ... & Coustau, C. (2005). Gene discovery and expression analysis of immune-relevant genes from *Biomphalaria glabrata* hemocytes. *Developmental & Comparative Immunology*, 29(5), 393-407.

Abou-El-Naga, I. F., & Mogahed, N. M. F. H. (2024). Immuno-molecular profile for *Biomphalaria glabrata*/*Schistosoma mansoni* interaction. *Developmental & Comparative Immunology*, 150, 105083.

Reviewer #4

(Remarks to the Author)

SUMMARY

=====

This is a review of the manuscript entitled "Immune targets for schistosomiasis control identified by a genome-wide association study of African snail vectors" (566873_0) by Pennance, Tennessen et al. In their manuscript, the authors identify loci involved in resistance to schistosome in the snail vector *Biomphalaria sudanica*, one of the most common vectors in Africa. They performed a genome-wide association study (GWAS) using snails derived from a field population and exposed to a natural population of schistosomes. Allele frequency comparison between infected and non-infected snails, followed use of amplicon-panel genotyping on individual snails, revealed two major genomic regions, *SudRes1** and **2**, acting additively. Interestingly, the authors also uncovered structure in their individually genotyped samples, revealing two ancestral populations. Ancestry was associated to the susceptibility/resistance phenotype and was accounted for in the analysis. Each genomic region showed enrichment in specific gene families, possibly linked to immune function. Susceptible and resistant alleles showed significant differences in terms of mutations and structural rearrangements, with possible balancing selection at play.

This work represents an impressive body of work and is relevant to both the field of schistosomiasis and, more broadly, the study of host-pathogen interactions. The approach and model used are strengths of the study. The methods and analysis are appropriate, though I have a few major comments detailed below. Overall, the manuscript reads well, with only minor typos and a few rewordings needed.

LEGEND

=====

- p.: page
- l.: line
- Fig.: figure
- Tab.: table

This review is written in [markdown](<https://en.wikipedia.org/wiki/Markdown>).

MAJOR COMMENTS

=====

The use of an alternative snail to *B. glabrata*, which has served as the host model for decades, to explore snail-schistosome compatibility is the most significant strength of the study. This approach has especially revealed that the genetic basis of host resistance is even more complex than previously thought, suggesting the involvement of specific loci in each system (BgGUA, BgBS90, Bs). This is critical to better understand how the system works. In addition, the authors used natural populations of snails and schistosomes which is the second major strength of the study, as it leverages the natural genetic diversity to fine-map QTLs (with some surprises such as the unexpected population structure). However, there are a few major points that need to be addressed.

Note: I did not have access to any of the datasets (S1-S6). I understand that this may be due to reasons beyond the authors' control. Some of the comments below may have been addressed by accessing these files.

Below the list of the major points:

- **Title:** Overall, I found the title misleading. The authors identified the genetic basis of host compatibility in *B. sudanica* by revealing two major regions that contain interesting candidate genes, which **may** be related to the snail immune system. As acknowledged by the authors, this could help pave the way for alternative strategies in schistosomiasis control as a long-term goal. However, the "immune targets" are not confirmed as being involved in immunity or actual causative genes,

and "schistosomiasis control" is beyond the scope of the present manuscript. I suggest that the authors find a more suitable title that reflects better the content of the manuscript.

- **Data and code sharing**:

+ **Genomic data**: The authors appropriately share their genome sequencing data and the code used for analysis, either by sharing scripts or providing sufficient information in the text (though providing a workflow would have been preferable for reproducibility). However, I suggest that they publish their code on a repository such as Zenodo or FigShare and provide an informative readme file describing the project's purpose and a brief description about the script usage. This is important because GitHub repositories can be moved or deleted by the owner or the company, potentially precluding access to the code in the future.

+ **Amplicon sequencing**: The authors seem to have shared the coordinates of the targeted sequences (dataset S3) and the genotypes obtained from sequencing (dataset S6). However, the sequencing data was not shared, preventing possible reanalysis. In addition, if the primer sequences are not proprietary, the authors should share them as well, to allow other groups to perform similar work.

+ **Admixture / PCA analysis / logistic regression**: no code or precise information has been shared for these analyses. The authors should correct that.

- **Methods and analyses**:

+ **Amplicon sequencing**: The Materials and Methods section provides little information on how the amplicon sequencing was performed, stating only that it was conducted by the GTseek company. If the company has a standard protocol, it should be cited. Otherwise information regarding the library preparation and the sequencing should be added, particularly regarding the amount of input DNA, the number of PCR cycles, the kits used, the number of sequencing batches (including the sample lists associated), the sequencing platform. In addition, there is limited information on how the data was processed and analyzed: How was sequencing quality control performed? Was there a threshold to remove bad reads? Were reads trimmed? Was a particular Phred score used?

+ **Admixture**: In addition to the limited information on how this analysis was performed, the interpretation of results when adding *B. choanomphala* raises questions, particularly regarding the absence of possible hybridization. I provide specific details below.

+ **Read depth normalization**: A read depth analysis was performed on the two genomic regions of interest but there is no details in the Material and Methods regarding this, especially normalization which is critical as variation in sequencing depth between libraries are very common.

+ **RNA-seq mapping**: This mapping was done using `bwa mem`. While this step was performed to confirm transcript prediction, the best practice would be to use a splice-aware aligner such as `STAR` to identify splicing sites and provide improved annotation. This is particularly critical with long-read data, which are likely to span introns. It is also important for the current study because the authors hypothesize that the identified structural changes in candidate genes might be responsible for the phenotype. So excluding technical artifacts here is important. Unless the authors have a justification for their choice, I suggest redoing the analysis with an appropriate tool.

+ **BLAST searches**: Several `blast` runs were performed but there is no information on parameters used in the Materials and Methods section for any of these runs. The authors should add this information.

- **Candidate genes**: The authors performed an extensive analysis of the two genomic regions linked to schistosome resistance. They identified interesting families of candidate genes and proposed working hypotheses for future work. However, this could be significantly strengthened by a small addition: Are these genes expressed in hemocytes, which are known to be the effector cells involved in resistance? I understand that, despite having performed RNA-seq on the *B. sudanica*, it is unlikely that the authors have tissue specific RNA data. However, there should be enough data on *B. glabrata* hemocytes to explore this (<https://doi.org/10.3389/fimmu.2022.956871>, <https://doi.org/10.1007/s00251-021-01236-3>, <https://doi.org/10.3389/fimmu.2018.02773><https://doi.org/10.1186/1471-2164-9-634>).

Comments on the text

- **I. 30**: "snail immunity": There is no clear demonstration of the role of the genes under the QTLs in the snail immunity. The genes are good candidates by their features but a formal demonstration is needed to claim that these two QTLs are related to immunity.

- **I. 98-99**: "1.5x coverage per snail (Dataset S1)": 1.5x per snail seems low and may have prevented the identification of other important regions. Another, maybe more relevant metric could be the read depth of coding regions (where the causative loci are likely located) because the snail genome contains many repeats, which may lower the alignment efficiency and the average coverage/read depth.

- **I. 128**: "($p < 1e-14$)": A p-value alone is meaningless. The authors should at least provide the test used. Access to the code written for analyzing the data would have been a plus.

- **I. 138**: "GWAS ancestry signal is not caused by interspecies hybridization": This section is confusing. The use of *B. choanomphala* is sensible; however, the result of the admixture analysis contradicts the authors' claim. If *B. choanomphala* belongs to group A, then the group A has *B. choanomphala* ancestry and hybridization is ongoing, with group B representing the ancestral *B. sudanica* population. However, the analysis itself seems strange. My understanding is that it was first done with *B. sudanica* samples from the field only, which led to the identification of the two groups, and

then the *B. choanomphala* and lab *B. sudanica* were assigned to these groups. However, this analysis should be performed using all samples at once to define populations and formally test for hybridization. In addition, the *B. choanomphala* sample in Fig. 2 is not apparent despite its mention in the caption.

- **l. 280**": "immune related genes" should be "genes possibly related to immunity". The genes highlighted have not been demonstrated to be involved in immune reaction in the current system.

In the Supplementary materials:

- **p. 4**": "We chose the model with the lowest AIC": what is the model chosen? The authors should add it.

Comments on the figures

- **Fig. S11**": How was the read depth normalized? Slight differences in sequencing depth between libraries could also explain the observation.

MINOR COMMENTS

=====

Comments on the text

- **l. 89 / 91**": "1400 F1" / "1,109 snails": the authors should be consistent in number notation. This should be checked throughout the text.

- **l. 125**": "(Fig. S5)": This figure is mentioned before Fig. S2. Same with Fig. S6 (l. 138) and Fig. S7 (l. 140). The authors should reorder supplementary figure accordingly to the order of their appearance in the text.

- **l. 264**": "confirmational" should be "conformational".

- **l. 276-277**": "The *SudRes1* region was ... in *B. sudanica* (27)": At first read, this sentence appears misleading as it suggests that *SudRes1* was already identified in ref. 27. The authors should rephrase this to avoid confusion.

- **l. 322**": "they do" should be "*B. choanomphala* does" if I understand correctly.

- **l. 335-346**": "The collection site ... lake areas": A map would be very appreciated to illustrate the text.

- **l. 359**": "a priori candidates": I suspect the authors refer to the PTC1 and 2 loci. The authors should cite them explicitly for helping the audience unfamiliar with the system.

In the Supplementary materials:

- **p. 2**":

+ "timed" should be "trimmed".

+ "MINLEN:50)" should be "MINLEN:50".

+ "(GCA_000237925.5" should be "(GCA_000237925.5)".

- **p. 3**": "Schistosoma 16S amplicon": It is unclear if the authors refer to the 16S rRNA gene of the schistosome mitochondrion or if this is a typo and refers to the 18S rRNA gene in the schistosome nuclear genome.

- **p. 4**": "if each allele has >10% the coverage of the other allele the genotype is designated heterozygote": This section is confusing to me. Are the authors considering the heterozygous state when allele A has 10 reads and allele B has 11 reads, or when allele A has 10 reads and allele B has 1 read? In the first case, what is the upper limit? In the second case, 10% seems very low to consider the individual heterozygous. Is there a rationale behind this threshold? In any case, the authors should clarify this section.

- **p. 7**": "The dotted lines ... no correlation.": I do not understand this sentence. The authors should rephrase it.

Comments on the figures

- **Fig. 1 & Fig. 3A**": characters from the y axis labels and the legend did not display correctly in Adobe Acrobat Reader v. 2024.005.20399.

- **Fig. 2**": The authors should place the letter C on the left of the figure for consistency and readability.

- **Fig. S2**": The authors should change the color palette chosen for the linkage groups as this is already used for designated A and B groups in Fig. 2 and is therefore confusing.

Version 1:

Reviewer comments:

Reviewer #1

(Remarks to the Author)

The revised manuscript has improved significantly. However, several important concerns regarding the rigor of the study remain and should be addressed.

Both the population structure analysis and the genotype-phenotype association analysis should account for familial relatedness among samples. While the population structure analysis has been revised using appropriate methods, familial relatedness was not considered in the genotype-phenotype association analyses (Fig. 2B–E). A conservative approach would be to restrict the association analysis to unrelated individuals. Alternatively, methods that incorporate a genetic relatedness matrix (GRM) into regression models could be employed.

In addition, the revised submission includes supplementary data that were not part of the initial submission. The dataset numbering does not follow the order of appearance in the manuscript. Furthermore, the use of commas as thousands separators is inconsistent within and across datasets and should be standardized.

Dataset S1

The number of reads mapping to *S. mansoni* is high, representing approximately 15-25% of total sequencing reads for both infection "negative" and "positive" pools. The proportions of *S. mansoni* reads also differ considerably between technical replicate pools. According to the methods section, *S. mansoni* reads were removed prior to mapping to the *B. sudanica* genome. However, the input read counts for *B. sudanica* alignment (Column AR) exceed the number of reads remaining unmapped to the *S. mansoni* genome (Column AC). Clarification is needed regarding this discrepancy. Moreover, there are multiple columns labelled "raw total sequences," the meaning of which is unclear.

Column A: No column header is provided.

Column B: SRA accession numbers would be preferable to file names.

Column C: The meaning of "Lane D" should be clarified.

Column E: The "file size (bytes)" field should be removed.

Columns F, K, O, U, AF, and AP: These columns appear redundant given the information provided in the header row.

Column S: Some cells contain missing data.

Dataset S2

Clarification is required regarding the assignment of contig_1504 and contig_351 to linkage group 1B. Although these contigs are linked, neither could be mapped to BioGlab47 Chromosome 1.

Dataset S3

P values greater than 1 are present in Column K and should be corrected. The genotype missing rate for individual markers should be reported. Allele 1 and allele 2 nucleotides and their counts for each population used in the association testing should be provided. A separate tab or dataset should be included to show the genotype matrix for individual *B. sudanica* samples (503 samples x 470 markers). This should include SRA accession numbers for each sample, along with relevant metadata such as ancestry proportions, infection status, and relatedness.

Datasets S4 and S5

The Excel files require formatting revisions. Some cells are colored, bolded, or highlighted. If these formats convey specific information, explanations should be provided, otherwise, the formatting should be standardized. Additionally, certain rows lack gene IDs and gene coordinates (i.e., both fields are marked as NA). Furthermore, Dataset S4 contains two tabs with redundant information, which should be consolidated or clarified.

Comments on newly included or revised figures:

Figure S2

Significance testing results should be included.

Figures S3 and S5

Definitions or descriptions should be provided for the terms “true positive,” “true negative,” and “false negative.” Alternatively, these labels should be revised for consistency with the terminology used in the main text and other figures.

(Remarks on code availability)

They provided the commands used in the analysis with detailed annotation. There are several perl scripts used for data processing, each performing simple tasks.

Reviewer #2

(Remarks to the Author)

(Remarks on code availability)

Reviewer #3

(Remarks to the Author)

The manuscript presents a genome-wide association study of African *Biomphalaria* snails, which serve as intermediate hosts for the parasite responsible for schistosomiasis in an endemic region of high transmission. I coincide with the authors that the recent revisions significantly enhance the quality of the manuscript without altering its primary conclusions. In fact, these modifications further corroborate the results indicating that the two genomic loci, *SudRes1* and *SudRes2*, are associated with resistance to *Schistosoma mansoni*.

I have examined the newly included PCA analysis and the Supplementary Information section titled “Validation of variants by amplicon sequencing.” These additions provide greater clarity, transparency, and reproducibility to the ancestry analysis, which I find creditable.

Upon reviewing the updated GWAS methodology, I gained a clearer understanding of the strategies implemented to incorporate ancestry as a covariate. I agree that accounting for ancestry through multiple regression analyses appropriately isolates genetic markers that have a meaningful effect on the phenotype.

Regarding the section titled “Evidence of a shifting snail population structure in Lake Victoria could lead to increased infections,” I verified that one of the publications I recommended has been duly cited in the discussion. I also appreciate the rationale for excluding Sire et al., recognizing its limited relevance in this context. In addition, I reviewed the revised Figure S7, which includes *B. choanomphala* in the ancestry-controlled PCA. The updated discussion more effectively addresses the observed cryptic population structure and the interspecies relationships among *Biomphalaria* snails in Lake Victoria.

The discussion of how ongoing admixture signatures may be influenced by the hydrologic patterns of Lake Victoria is now more nuanced and scientifically compelling. I agree with the authors that these findings highlight the substantial variation in pathogen resistance between closely related snail populations. This variability likely contributes to the persistent schistosome transmission in lake regions harboring highly susceptible *Biomphalaria* populations.

I also examined the new comparison in the discussion section between single-cell RNA sequencing datasets from *Biomphalaria glabrata* hemocytes and the study’s findings. This interspecies comparison greatly enhances the contextualization of the results. I concur with the authors that incorporating this perspective enriches the study, while the observed substantial differences between *B. sudanica* and *B. glabrata* justify limiting an extensive review of the latter in this manuscript.

Furthermore, I confirmed that the figures now correctly display the axes, enhancing the accuracy of data presentation.

Overall, I approve of this revised version of the manuscript and have no further suggestions or corrections.

(Remarks on code availability)

I confirmed that the code associated with the bioinformatic analysis has been made publicly available in a repository. Upon reviewing the scripts, I found them to be clearly written and fully reproducible. Even the manually created scripts are completely shared, ensuring transparency. Researchers wishing to replicate the analysis should have no difficulty doing so. Additionally, I verified that the code is logically sound, as the bioinformatic tools and overall pipeline adhere to standard practices commonly used in genetic data analysis.

Reviewer #4

(Remarks to the Author)

This is the second review of the manuscript now entitled “Genes linked to schistosome resistance identified in a genome-

wide association study of African snail vectors" (566873_0) by Pennance, Tennessen et al.

The authors adequately addressed the comments and implemented the necessary changes.

There are a few typos in the main text:

- **l. 320**": "identified" should be "identified" (i missing).
- **l. 321**": "Consistency" should be "Consistently" (l missing).

There is one typo in the supplementary methods:

- **p. 4**": "due their ambiguous" should be "due to their ambiguous".

(Remarks on code availability)

While a complete workflow would have been appreciated to facilitate the reproducibility, the code, comments and usage messages are enough to rerun the analysis.

Version 2:

Reviewer comments:

Reviewer #1

(Remarks to the Author)

The revised manuscript is much improved and I have no farther concerns. I am pleased with the revision and fully support moving forward with its acceptance.

(Remarks on code availability)

Reviewer #2

(Remarks to the Author)

(Remarks on code availability)

made.

Summary:

We appreciate the work of the four reviewers and found their insights helpful in strengthening our manuscript. We have substantially revised our manuscript in response to their comments, and we believe it is now meaningfully improved. In particular, we have redone our validation analyses to better account for ancestry, relatedness, and the effects of deletions and duplications on genotyping accuracy, as suggested. These adjustments do not qualitatively change our conclusions, as markers in both *SudRes1* and *SudRes2* are still significantly validated. We agree with the reviewers that this new analytical approach is more appropriate, and therefore we have changed the manuscript, including Figure 2, to reflect these methods. We find it encouraging that our results are robust to the details of which analysis methods are used, strengthening our confidence in them.

We also have included many more details on our methods, as suggested. Our supplemental methods section is now much longer, and the number of supplemental figures has increased by four. We have also made the relevant code and additional data (most notable the amplicon panel sequencing data) available online that was previously not shared. Other changes improve clarity or provide new information to support our conclusions, such as the validation of *SudRes2* in our linkage cross F2s.

Specific responses to reviewer comments are listed below in **bold blue text** in a point by point fashion. We also include in our submission the manuscript and supplementary data with and without tracked changes.

Yours sincerely,

Michelle Steinauer (on behalf of all co-authors)

Reviewer #1 (Remarks to the Author):

The manuscript addresses the urgent need for innovative strategies to control schistosomiasis, particularly by manipulating snail vector immunity. Through a genome-wide association study (GWAS) of African *Biomphalaria sudanica* populations, the authors identified two genomic loci, *SudRes1* and *SudRes2*, linked to resistance against *Schistosoma mansoni*. These loci encode receptor-like protein tyrosine phosphatases and leucine-rich repeat-containing G-protein-coupled receptors, highlighting immune functions relevant to host-pathogen interactions. While this study represents a significant advancement in the field, several concerns must be addressed to ensure that its conclusions are well supported. If confirmed through more rigorous analysis, these findings could serve as a foundation for identifying genetic targets for vector control and advancing research on vector-based schistosomiasis interventions.

We thank the reviewer for highlighting the importance of our work, and for the suggestions to improve the rigor of our analysis. As described below, we have revised our analyses accordingly and strengthened our conclusions.

Major concerns:

1. To perform experimental *S. mansoni* infections (phenotyping) and subsequent genotyping experiments, 1,400 F1 snails derived from 329 parental snails were used. Even if all F1s were generated through cross-fertilization of unrelated parental snails, this experimental design would still result in the sampling of closely related siblings. However, neither the population structure analysis nor the genotype-phenotype association analysis adequately accounted for this familial relatedness among samples. Both PCA and ADMIXTURE analyses assume that samples are unrelated, and these methods for inferring genetic ancestry can fail when applied to related samples (PMID: 25810074). We strongly encourage the authors to use methods specifically developed for analyzing samples with familial relatedness, such as PC-AiR (<https://bioconductor.org/packages/devel/bioc/vignettes/GENESIS/inst/doc/pcair.html>). After identifying the subset of unrelated individuals in the dataset, PCA and ADMIXTURE analyses can be run using them as a reference set, while the remaining samples (which may be related to the reference set) can then be projected onto the resulting population structure.

This is an excellent suggestion. We have now incorporated PC-AiR into our analysis to account for relatedness. As suggested by the reviewer, we then ran ADMIXTURE on the subset of samples identified by PC-AiR as unrelated, and subsequently projected the remaining samples onto the ADMIXTURE results. The resulting ancestry estimates were then controlled for in our validation analyses. These ancestry estimates are highly correlated with our previous estimates and do not qualitatively change our results, but we agree with the reviewer in favoring this new approach.

2. PCA (Fig. S5) shows that PC2 (10% of the variance), rather than PC1 (42% of the variance), represents the ancestry component identified by ADMIXTURE (Fig. 2A). This indicates the presence of an orthogonal genetic principal component (PC1) with a higher variance contribution that cannot be attributed to the ADMIXTURE ancestry used in the regression modeling. This discrepancy raises concerns about the robustness of ancestry inference and must be addressed to ensure the validity of the downstream association analysis. It could be useful to inspect the PC loadings to ensure that they broadly represent variation across the genome, rather than one or a small number of genomic regions. Additionally, including a PCA plot with phenotype labels (i.e., resistant or susceptible), alongside ancestry proportions, could improve data interpretation.

This is another important point raised by the reviewer. When we generate a PCA using PC-AiR, the ADMIXTURE-identified ancestry is now captured by PC1 (Fig. S5). So we suspect that our previous PCA may have been compromised by not properly accounting for relatedness. We no longer show the old PCA, and just show PCAs generated with PC-AiR (see also Fig. S7). We also show that ADMIXTURE models with higher K values have higher Cross-validation errors (Fig. S6), so our K=2 model is the appropriate one to use to account for ancestry.

3. Amplicon-seq data quality control (QC) needs to be more transparent and rigorous. Inclusion/exclusion thresholds for missingness per marker and per individual were overly lenient and inconsistently applied, varying substantially between analyses without clear justification. These thresholds as either minimum or maximum proportions should be reported, as appropriate,

to improve transparency in both the main text and the supplementary material (e.g., "<10% missing" instead of the number of genotypes/individuals). Including additional QC plots for the amplicon data, similar to Fig. 4 in PMID: 25476721, would help readers assess data quality.

We have clarified our genotyping approach. We now include a supplementary figure (Fig S4) modeled on Fig. 4 in PMID: 25476721, as suggested. The overall 2% missing data per locus on the amplicon panel (discussed in more detail below) was not a threshold we chose directly, but simply a consequence of the criteria we used for calling genotypes, as shown in this figure. For the missingness thresholds we did choose, we have modified them to be more consistent throughout the paper, and to be framed as percent missing rather than the number of non-missing genotypes as suggested by the reviewer. For example, in both our validation regression analysis (Fig 2C) and our analysis that looks at Group A or Group B separately (Fig 2B), we now require <20% missing genotypes at a locus. Arguably this is still quite lenient, but we reason that missing samples can be safely ignored as if they had never been genotyped in the first place, since what matters more is whether the number of good samples is large enough for statistical power to find an association. Sometimes missing data might be telling us something about the inheritance patterns at the locus (e.g. null alleles), but we don't need to discard loci from analysis for this reason (discussed more below).

4. Two resistance-associated regions (SudRes1 and SudRes2) were identified, and both regions exhibit complex structural variations (Figs. 3 and 4). When SNPs overlap with CNVs (both deletions and duplications), the accuracy of genotype and allele counts may be compromised, potentially leading to spurious genotype-phenotype associations. Liu et al. (PMID: 30019117) clearly demonstrate in detail how the coexistence of CNVs and SNPs in the same genomic region can distort significance calculations in association studies. Importantly, this issue is not limited to pool-GWAS analysis; amplicon analysis of individual snails is also affected. For example, heterozygous deletions lead to a hemizygous allele at corresponding genomic regions, and these can be taken as homozygous, distorting their allele counts. The significance of SudRes1 and SudRes2 was largely supported by four SNPs, all of which overlap with CNVs. The authors generated PacBio and Illumina whole-genome sequencing data from individual snails and upon examination of their raw read alignments for evidence of CNV colocalization with significant SNPs, the following IGV screenshots were generated that display the samples in this order: Bsud111, Bs2280, BsKEMRI, Bs163, Bs110, and Bs5-2.

(1) c6:3,490 (most significant SudRes1 variant) colocalizes with a deletion CNV (Bs163).

(2) c582:65,596 (representative SudRes1 variant) colocalizes with a deletion CNV (Bs2280 and Bs163)

(3) sc94:2,166,296 (most significant SudRes2 variant) colocalizes with a duplication CNV (Fig. 4C). This SNP appears as a heterozygous variant in Bs2280 with ~2x coverage because reads originating from two paralogous regions are aligned to the same locus.

(4) sc94:2,174,117 (representative SudRes2 variant) colocalizes with a duplication CNV (Fig. 4C). Although this SNP appears as a homozygous variant in Bs2280, reads originating from two paralogous regions are aligned to the same locus (2x coverage). This duplication can be seen

clearly in Fig. 4C. Sequence variations between paralogous copies may confound genotyping in this region.

The authors need to demonstrate that the association signals in *SudRes1* and *SudRes2* are supported by SNPs (in high LD with these regions but) located outside of CNVs and that they are not artifacts caused by distortions in allele/genotype counts due to the coexistence of CNVs and SNPs in the same genomic region. The authors could first identify CNV regions in the genome using WGS data and assembly comparisons and exclude these regions from SNP-based association analysis.

This is a critical point that we did not fully appreciate in our original analysis, and we thank the reviewer for bringing it to our attention. As explained by Liu et al. 2018, CNVs can cause errors when hemizygous SNPs are incorrectly called as homozygous. CNVs certainly do occur in *SudRes1* and *SudRes2* as the reviewer points out, evident in Figures 3 and 4.

This is not an issue for the pooled sequencing, since we don't call genotypes and simply look at allele frequencies (the issue of uneven coverage is discussed below). However, it is an issue for our amplicon genotypes. Previously, we fit additive regression models to our amplicon genotype data, counting the number of alleles as 0 (homozygous), 1 (heterozygous), or 2 (alt homozygous). This approach can be compromised by CNVs. To fix this, we now model all markers as dominant under a binary classification scheme: 0 (no copies of a given allele) or 1 (at least one copy of that allele). For each marker, we consider that either allele could have a dominant effect on phenotype and test the marker both ways. This approach is agnostic to CNVs, since it makes no assumptions about heterozygosity versus hemizyosity. If samples with at least one copy of an allele are significantly more/less likely to be infected, we can conclude there is a real phenotype-genotype association, whether it is a true Mendelian locus or a construct impacted by deletions or duplications. This new method changes the results slightly, as shown in the updated Figures 2C and 2E, but does not change our conclusions. *SudRes1* and *SudRes2* are still significantly validated. We now make it clear in the manuscript methods why we have adopted this approach and what the CNV-related caveats are.

Also note that for both *SudRes1* and *SudRes2*, we finalize our model (Fig 2E) using representative markers (c582:65,596 and sc94:2,174,117) that show largely Mendelian segregation in our GWAS data (Fig 2D; all three genotypes are observed in reasonable proportions, even though we cannot rule out some influence by deletions or duplications). We now make it clear (Fig S9) that there is large deletion across much of *SudRes1* in Bs163, as identified by the reviewer, though this deletion does not prevent us from detecting segregating variants within non-deleted genotypes, which can be modeled with dominance effects.

We appreciate the suggestion to identify markers in CNV-free portions of *SudRes1* and *SufRes2*, but this is not practical. Developing a new amplicon panel, or alternate means of genotyping hundreds of samples, is beyond the scope of this paper. The density of CNVs in these genomic regions is high and incompletely characterized since our assemblies are only partial, so it may not be possible to find reliable Mendelian markers close to the GWAS

outliers. In any case, additional genotyping data is unnecessary because our binary dominance analysis is robust to potential CNV artifacts.

Minor concerns:

L100: Correlation between technical replicate pairs were assessed using Fisher's exact test p values (Fig.S1). This analysis ignores the direction of allele frequency differences. Correlating allele frequencies between replicates would be more informative.

This is a good suggestion. We have replaced this figure with a new figure (new Fig S2) showing differences in allele frequency (between positive and negative pools) between replicates. It is clear that the technical replicates are correlated.

L103: "In the pooled-GWAS results, genotype-phenotype association p values ranged as low as $1e-30$, including 45 variants (0.001%) with $p \leq 1e-15$ and 1,930 variants (0.04%) with $p \leq 2.5e-9$. (Fig. 1)." When Fisher's exact test is used in pool-seq analysis, read counts are used in place of true allele counts, which cannot be directly measured. Because 493 positive and 295 negative snails were pooled, the total ploidy in each pool was 986 and 590, respectively. When sequencing depth exceeds these ploidy counts, pool-seq p values may be lower than the true significance of allele counts differences. In such cases, p values cannot be simply interpreted as indicators of association strength, as they become influenced by sequencing depth (i.e., higher depth results in lower p values even when allele counts remain unchanged). This dependency of Fisher's exact test p values on sequencing depth also confounds the comparison of association signals between loci with unequal read depths in GWAS. Genomic read coverage fluctuates along chromosomes due to random distribution and biases such as GC content where high GC regions exhibit elevated coverage. Fisher's exact test is more likely to detect allele frequency differences in high coverage regions. One way to mitigate this bias is to subsample to a uniform coverage by randomly sampling bases before estimating the significance of allele frequency differences (PMID: 22025480).

This is another excellent insight by the reviewer. We have now re-run our pooled GWAS analysis while controlling for coverage (Fig S3). Specifically, we subsampled the GWAS data to 300 True Negative reads and 300 True Positive reads (sites failing to meet this depth were discarded). Variants in *SudRes1* and *SudRes2* are still among the top outliers, so they would have been selected for the amplicon panel even if we had analyzed the data in this manner from the start. Other outliers do not change substantially in relative ranking.

We want to stress that the pooled GWAS results do not indicate biological significance, for the reasons noted by the reviewer and due to other potential artifacts. The p-values are not taken to be tests of a hypothesis, but rather a means of identifying candidates for the amplicon panel. It is only by independently validating the loci that we have confidence in their phenotypic association. Thus, if false positives were identified in the pooled GWAS and included on the panel (indeed, many were!), it will not matter since the conclusive evidence is in the validation.

L115: Methods section show describe how these 201 ‘neutral’ markers were selected. Using the amplicon panel (which also contains *S. mansoni* targets), a linkage map was constructed by crossing parental snails that differ in their susceptibility to *S. mansoni* infection. Was this amplicon dataset used for pedigree-based mapping of resistance loci?

Yes, we challenged the linkage cross offspring with schistosomes and classified them as positive or negative, as with the GWAS snails. Due to our relatively small sample size, we did not have the power to conduct a formal QTL analysis, which is why we initially left this out of the manuscript. However, for full transparency we have now added a figure showing *SudRes1* and *SudRes2* in the linkage cross (Fig S9). *SudRes1* has a null allele in one parent and is not significantly correlated with phenotype, either because of low statistical power or because the resistance allele is not present in the cross. However, *SudRes2* is significantly correlated with infection, in the same direction as the GWAS and validation snails. Thus, the cross provides additional evidence supporting this genotype-phenotype association.

L124: “The amplicon panel data (of which median missing data was 2% per locus and 2% per individual) revealed a signal of population structure.” Instead of reporting the median missing rate, reporting of the minimum/maximum missing rates could be done for transparency. Were the inclusion/exclusion thresholds sufficient to ensure a robust analysis? A case/control nonrandom missingness test is often performed in GWAS. Fisher's exact test can be applied to missing call counts at each variant to check for systematic biases in the dataset.

We now include more details on missingness for transparency, detailed in the results section “*Amplicon panel genotyping reveals population structure and validates variants associated with resistance*”. Missing genotypes per sample ranged from 0.2% to 100% (only a single sample was 100% missing), with median 1.7% missing. 90% of samples had between 0.6% and 6.0% missing genotypes. Missing genotypes per locus ranged from 0.2% to 78%, with median 1.6% missing. 90% of loci had between 0.8% and 14.3% missing genotypes. We also ran Fisher’s exact tests on missingness for every locus to test for association with infection phenotype. Only 12 markers showed $P < 0.05$, fewer than expected by chance for 469 markers, with six showing more missingness in infected snails and six showing more missingness in uninfected snails. The only marker with $P < 0.001$ was c415:859044, one of the worst-performing markers with 78% missingness, which we do not expect had a meaningful effect on our analysis. Thus, we do not see systematic biases.

We note that in some cases missing genotypes may reflect real genotypic information (i.e. null alleles) rather than PCR failure, but we conservatively treated them as missing rather than trying to link them to phenotype (except for *SudRes1* in the linkage cross, see Fig S9), and our new binary analysis approach should be robust to null alleles.

L126: “With $K=2$ ancestral populations ($CV = 0.53$, versus 0.61 for $K=1$)..” What were the CV values for $K>2$? This information could be included in a supplementary figure. In addition, ADMIXTURE can estimate parameter standard errors using bootstrapping and they can be displayed in Fig. 2A.

We have added a supplementary figure showing ADMIXTURE results at higher K values (3 and 4). K of 2 has the lowest CV so we treat it as the best estimate. We have also added standard errors inferred from bootstrapping to the ancestry plot Fig 2A.

L135: This approach is likely not as robust as PCA or ADMIXTURE. Once genetic structure is modelled in ADMIXTURE using unrelated individuals, *B. choanomphala* and *B. sudanica* inbred lines can be "projected" onto the population structure to estimate their ancestry. The supplementary information mentions that a PCA analysis was performed including these samples, but the result is not included in the manuscript.

This is a good point. We have now generated a PCA using PC-AiR (Fig S7) that incorporated the GWAS snails along with reference samples of *B. choanomphala* and *B. sudanica*. This shows that the ancestry component in our GWAS is not the same as the choanomphala-sudanica difference.

L141: It is questionable whether the analysis presented in Fig. 2B adequately accounted for ancestry, as ancestry proportions vary continuously (Fig. 2A). Individual PCA plots for Group A and Group B with phenotype labels could help determine whether this approach effectively accounted for ancestry.

We agree with the reviewer. The ancestry component varies continuously from 0% to 100%, so if we divide the snails into two groups of >50% or <50%, there is still a lot of ancestry variation within these groups that remains unaccounted for. Therefore, we changed our analysis in Figure 2B by only considering snails with <10% or >90% ancestry of Population 2 (see groups in Fig2A). All admixed snails between 10-90% are ignored. This doesn't completely get rid of the ancestry effect, but it greatly minimizes it. Remarkably, even though this new analysis has a substantially reduced sample size (n = 46 and 34 for Groups A and B respectively), we still validate two variants. Specifically which variants are now validated has changed slightly, but they are still in *SudRes1* as before. This does not change the analysis in Fig 2C, which fully accounts for ancestry by treating it as a continuous variable.

L147: "acted as dominant markers (only two genotypes observed)". It is highly likely that only two genotypes were observed not because these variants acted as dominant markers but due to a genotyping error in which paralogous alleles were amplified by primers, resulting in the variants being incorrectly genotyped as heterozygotes. This must be clearly conveyed to the readers in the main text.

We have clarified our language in the text by changing to "acted as dominant markers (only two genotypes observed, possibly due to co-amplification of paralogs)". By dominant markers we mean exactly the scenario the reviewer describes. Two different loci are amplified simultaneously, and the SNP occurs on only one of these two loci, so both heterozygotes and true homozygotes appear in our data to be heterozygotes. Dominance means that heterozygotes are indistinguishable from homozygotes of a particular allele. We now explain this in detail.

L428: Datasets S1–S6 were not accessible to reviewers and need to be made available. The genome assembly of Bs2280 is also not available on Figshare.

Our apologies, the Datasets S1-S6 were missed from the files sent to the reviewers, but are now included in the resubmission. The Bs2280 genome assembly is now publicly available on FigShare, as it seems our private link (for use at this review stage) did not work for the reviewers.

Reviewer #2 (Remarks to the Author):

We thank reviewer 2 for contributing to the other reviewer comments that have helped to significantly strengthen this manuscript.

Reviewer #3 (Remarks to the Author):

The manuscript describes a genome-wide association study of African *Biomphalaria* snails that are intermediate hosts of the parasite that causes schistosomiasis in an endemic area of high transmission. The study identified two genomic regions that were significantly associated with snail immunity. The significant aspect of this work is that it is the first time an exhausted genome-wide study has been done in a wild population of *Biomphalaria sudanica* snails with substantial statistical power since 493 infected and 295 uninfected snails were sequenced for this work. Furthermore, the study employs the most recent discoveries and advancements in genomics of this snail genus enhancing the understanding of host-parasite interactions in natural populations.

I read carefully the material and methods information provided with the manuscript, and I did not find any flaws in the study. Moreover, I found the approach very original, since it is very hard to obtain high-quality sequencing coverage per individual snail, therefore the GWAS-pooled approach is an efficient method to assess single nucleotide polymorphisms in a wild population. Furthermore, the authors also employed another genotyping technique that allowed them to validate their findings in the pooled sequencing approach which provides robustness to their data.

Besides the authors include in the discussion the limitations of their results highlighting that the two uncharacterized genomic regions associated with resistance to *S. mansoni* infection have low read coverage and few annotated genes.

I agree with the authors that this work represents an important step toward molecular-informed vector control to combat schistosomiasis in high-transmission regions and highlights the importance of the genetic background and environmental factors on the impact of resistance alleles.

While the data and overall methodology presented in the study are clear, I found the methods related to ancestry somewhat difficult to follow. Since not all readers may be familiar with population structure analysis, it would be helpful if the authors provided a more detailed explanation in the methodology section, specifically regarding how snail ancestry and schistosome resistance were evaluated. Specifically, in the paragraph where they mention: “We assessed population structure in genotyped-validation snails and genotyped-pooled-GWAS snails using ADMIXTURE (15) and principal component analysis (PCA) in R”, additional details are needed.

We have re-done the PCA and ADMIXTURE analysis in response to the comments of reviewer 1, and we now provide more details about this in the Supplementary Information Methods section “Validation of variants by amplicon genotyping”. We believe this will help make the ancestry piece more clear to readers.

While Figure S5 displays three different subpopulations of snails, it is unclear why the samples are colored based on population 2 ancestry. It would be helpful if the authors could explain the significance of Population 2 in this context. Is this population used as a reference population?

We have re-done our PCA analysis in response to reviewer 1 and removed this figure. Our initial goal was to show that PCA results and ADMIXTURE results are highly correlated. We now show this directly in a new figure (Fig S5) by plotting PC 1 from the PCA against Population 2 ancestry estimate from ADMIXTURE. Population 1 and Population 2 are the two ancestral populations inferred by ADMIXTURE (the 1-2 numbering is arbitrary). There is no previous literature on those populations as such, since this ancestry pattern is a new result of our paper. In this figure and elsewhere we refer to Population 2 ancestry as a metric, but we equally could have chosen to frame things with respect to Population 1 ancestry, since they are each other’s opposites and sum to one in each snail.

Moreover, can the authors clarify how they validated that the variants identified in the GWAS analysis are specifically associated with snail immunity, rather than reflecting genetic differences between distinct population groups?

Providing this information would improve the transparency and reproducibility of the study.

By including ancestry as a factor in our analysis, we can account for it and thus test if these loci are associated with phenotype even beyond this ancestry effect. We did this several ways. In Fig 2B, we restrict analysis to snails with nearly 100% ancestry (>90%) from one population or the other (Group A or Group B), and we look for genotype-phenotype associations within these ancestry-homogenized groups. In Fig 2C, we include ancestry as a variable in multiple regression, so a genetic marker will only be significant if it has an effect on phenotype that isn’t already captured by the ancestry variable. In Fig 2E, we test the effects of ancestry and both validated loci all together. This is a critical aspect of our analysis, and we have revised our wording to make it clearer.

The statistical methods and analytical approach in this manuscript are robust, and I do not have any significant suggestions for improvement. I also believe that no additional experiments or data are necessary for the study.

There are only a few minor adjustments to consider:

Figure 1 has an issue with the y-axis values that should be corrected.

Similarly, Figure 3A has the same issue with the y-axis values and requires adjustment.

We have resolved this issue by providing Figure 3 (and all other figures) in a .png format inserted in the text, in addition to the original PDF format submitted. We will ensure that the figure axes are displayed correctly in the final manuscript.

The results are presented with substantial context and thoughtful integration of previous studies. However, the discussion could be further strengthened by including literature on cryptic population structures observed in other *Biomphalaria* species. This would provide additional insights and help hypothesize the causes of the cryptic population structure identified in this study. Some suggestions include:

Standley, C. J., Goodacre, S. L., Wade, C. M., & Stothard, J. R. (2014). The population genetic structure of *Biomphalaria choanophala* in Lake Victoria, East Africa: implications for schistosomiasis transmission. *Parasites & vectors*, 7, 1-10.

Sire, C., Langand, J., Barral, V., & Theron, A. (2001). Parasite (*Schistosoma mansoni*) and host (*Biomphalaria glabrata*) genetic diversity: population structure in a fragmented landscape. *Parasitology*, 122(5), 545-554.

Regarding the cryptic population structure and interspecies relationships of *Biomphalaria* in Lake Victoria mentioned by the reviewer, and in reference to adding the reference by Standley et al. (2014), we have now addressed this point more directly in our study by, for example, including *B. choanophala* data in our ancestry controlled PCA (Fig S7). We add note to this, including reference to the Standley et al. study, in the discussion section “Evidence of a shifting snail population structure in Lake Victoria could lead to increased infections”. Since the Sire et al. study refers to South American species *B. glabrata*, and diversity within geographically isolated populations and not in a physically connected environment such as those in Lake Victoria discussed here, we did not include this citation.

The manuscript would also benefit from the inclusion of additional references on immune mechanisms in laboratory populations of *Biomphalaria glabrata* mentioned in line 355. Specifically, citing studies that explore genomic regions associated with immune responses to schistosome parasites, as well as research on *Biomphalaria*-*Schistosoma* molecular interactions, would provide further support and enhance the contextualization of the findings.

A few suggestions are:

Mitta, G., Galinier, R., Tisseyre, P., Allienne, J. F., Girerd-Chambaz, Y., Guillou, F., ... & Coustau, C. (2005). Gene discovery and expression analysis of immune-relevant genes from *Biomphalaria glabrata* hemocytes. *Developmental & Comparative Immunology*, 29(5), 393-407.

Abou-El-Naga, I. F., & Mogahed, N. M. F. H. (2024). Immuno-molecular profile for

Biomphalaria glabrata/Schistosoma mansoni interaction. Developmental & Comparative Immunology, 150, 105083.

The reviewer is referring to the section ‘Evolutionary dynamics of snail-schistosome interactions’, and suggests to add additional referencing. Following comments from reviewer 4 to add analysis related to hemocyte expression data, we have now added a sentence to the discussion which makes comparisons to single cell RNA seq datasets from *B. glabrata* hemocytes, which provides some further interspecies comparisons of the *B. glabrata* system and *S. mansoni* interactions. We hope the reviewer agrees that this addition adds some further contextualization of our study . We also want to be clear in our discussion that we are finding different things in *B. sudanica* in comparison to *B. glabrata*, justifying why our work focusses on the African species and why a longer review of the *B. glabrata* literature in relation to the snail-schistosome interactions (many of which are already included in the article) is not needed here.

Reviewer #4 (Remarks to the Author):

SUMMARY

=====

This is a review of the manuscript entitled "Immune targets for schistosomiasis control identified by a genome-wide association study of African snail vectors" (566873_0) by Pennance, Tennesen et al. In their manuscript, the authors identify loci involved in resistance to schistosome in the snail vector *Biomphalaria sudanica**, one of the most common vectors in Africa. They performed a genome-wide association study (GWAS) using snails derived from a field population and exposed to a natural population of schistosomes. Allele frequency comparison between infected and non-infected snails, followed use of amplicon-panel genotyping on individual snails, revealed two major genomic regions, *SudRes1** and **2**, acting additively. Interestingly, the authors also uncovered structure in their individually genotyped samples, revealing two ancestral populations. Ancestry was associated to the susceptibility/resistance phenotype and was accounted for in the analysis. Each genomic region showed enrichment in specific gene families, possibly linked to immune function. Susceptible and resistant alleles showed significant differences in terms of mutations and structural rearrangements, with possible balancing selection at play.

This work represents an impressive body of work and is relevant to both the field of schistosomiasis and, more broadly, the study of host-pathogen interactions. The approach and model used are strengths of the study. The methods and analysis are appropriate, though I have a few major comments detailed below. Overall, the manuscript reads well, with only minor typos and a few rewordings needed.

LEGEND

=====

- p.: page

- l.: line
- Fig.: figure
- Tab.: table

This review is written in [markdown](<https://en.wikipedia.org/wiki/Markdown>).

MAJOR COMMENTS

=====

The use of an alternative snail to *B. glabrata*, which has served as the host model for decades, to explore snail-schistosome compatibility is the most significant strength of the study. This approach has especially revealed that the genetic basis of host resistance is even more complex than previously thought, suggesting the involvement of specific loci in each system (BgGUA, BgBS90, Bs). This is critical to better understand how the system works. In addition, the authors used natural populations of snails and schistosomes which is the second major strength of the study, as it leverages the natural genetic diversity to fine-map QTLs (with some surprises such as the unexpected population structure). However, there a few major points that needs to be addressed.

Note: I did not have access to any of the datasets (S1-S6). I understand that this may be due to reasons beyond the authors' control. Some of the comments below may have been addressed by accessing these files.

Apologies for the inconvenience, the supplementary datasets 1-6 were missed from the version sent to reviewers but should be included in this resubmitted version.

Below the list of the major points:

Title: Overall, I found the title misleading. The authors identified the genetic basis of host compatibility in *B. sudanica* by revealing two major regions that contains interesting candidate genes, which **may** be related to the snail immune system. As acknowledge by the authors, this could help pave the way for alternative strategies in schistosomiasis control as a long-term goal. However, the "immune targets" are not confirmed as being involved in immunity or actual causative genes, and "schistosomiasis control" is beyond the scope of the present manuscript. I suggest that the authors find a more suitable title that reflects better the content of the manuscript.

The title of the submitted manuscript is: “Immune targets for schistosomiasis control identified by a genome-wide association study of African snail vectors”, and underlines are the parts highlighted by the reviewer as being potentially misleading. We have therefore generated a simplified title for the manuscript in the revised submission: “Genes linked to

schistosome resistance identified in a genome wide association study of African snail vectors”

- **Data and code sharing**:

+ **Genomic data**: The authors appropriately share their genome sequencing data and the code used for analysis, either by sharing scripts or providing sufficient information in the text (though providing a workflow would have been preferable for reproducibility). However, I suggest that they publish their code on a repository such as Zenodo or FigShare and provide an informative readme file describing the project's purpose and a brief description about the script usage. This is important because GitHub repositories can be moved or deleted by the owner or the company, potentially precluding access to the code in the future.

As well as the genomic data uploaded to NCBI (BioProject: PRJNA1149315) and FigShare (figshare.com/projects/Genome_Wide_Association_Study_of_Biomphalaria_sudanica/230615) project pages, we have additionally added the command line, Perl and R code/scripts used in the analysis to the FigShare page.

+ **Amplicon sequencing**: The authors seem to have shared the coordinates of the targeted sequences (dataset S3) and the genotypes obtained from sequencing (dataset S6). However, the sequencing data was not shared, preventing possible reanalysis. In addition, if the primer sequences are not proprietary, the authors should share them as well, to allow other groups to perform similar work.

The primer sequences are in dataset S3. We apologize that this file was not accessible previously. In addition we have uploaded all sequences from the amplicon panel sequencing to the NCBI Sequence Read Archive (BioProject: PRJNA1149315, Accessions: SRR32947997 - SRR32948590) within the same BioProject as the GWAS sequence data.

+ **Admixture / PCA analysis / logistic regression**: no code or precise information has been shared for these analyses. The authors should correct that.

We have now added more details on these analyses to the supplementary methods. Code for *perl* and *R* scripts generated in this project are shared in the FigShare Project: www.figshare.com/projects/Genome_Wide_Association_Study_of_Biomphalaria_sudanica/230615

- **Methods and analyses**:

+ **Amplicon sequencing**: The Materials and Methods section provides little information on how the amplicon sequencing was performed, stating only that it was conducted by the GTseek company. If the company has a standard protocol, it should be cited. Otherwise information regarding the library preparation and the sequencing should be added, particularly regarding the amount of input DNA, the number of PCR cycles, the kits used, the number of sequencing batches (including the sample lists associated), the sequencing platform. In addition, there is limited information on how the data was processed and analyzed: How was sequencing quality

control performed? Was there a threshold to remove bad reads? Were reads trimmed? Was a particular Phred score used?

We now explain our genotyping methodology in more detail. See Fig. S4 showing how we defined homozygotes and heterozygotes. The genotyping service provided by the gtseek company provides counts of all alleles rather than raw reads, so we are unable to trim or filter reads. It is clear that the data are somewhat noisy, and homozygotes will often have a small percentage of alternate alleles. Thus, we devised a conservative approach that discarded low-coverage genotypes. At higher coverage, homozygotes and heterozygotes form distinct clusters that can be readily distinguished.

+ ****Admixture****: In addition to the limited information on how this analysis was performed, the interpretation of results when adding *B. choanomphala* raises questions, particularly regarding the absence of possible hybridization. I provide specific details below.

See our new PCA that shows *B. choanomphala* in relation to our GWAS samples (Fig S7).

+ ****Read depth normalization****: A read depth analysis was performed on the two genomic regions of interest but there is no details in the Material and Methods regarding this, especially normalization which is critical as variation in sequencing depth between libraries are very common.

We now explain our methods in full detail. We revised the depth figures for *SudRes1* and *SudRes2* (now Figs S14 and S17) to show normalized depth (previously it showed raw unadjusted depth) We have also added a new analysis (Fig S3) in which we fully control for read depth.

+ ****RNA-seq mapping****: This mapping was done using `bwa mem`. While this step was performed to confirm transcript prediction, the best practice would be to use a splice-aware aligner such as `STAR` to identify splicing sites and provide improved annotation. This is particularly critical with long-read data, which are likely to span introns. It is also important for the current study because the authors hypothesize that the identified structural changes in candidate genes might be responsible for the phenotype. So excluding technical artifacts here is important. Unless the authors have a justification for their choice, I suggest redoing the analysis with an appropriate tool.

Thanks to the reviewers comment on this. We agree that STAR is the appropriate aligner to use for transcript alignment to a whole genome where splice sites are a factor to consider, but in this instance we are only confirming that the predicted gene (not annotated in the published *B. sudanica* genome likely due to variation) transcripts, which contain no splice sites, were present in transcript data we have available from *B. sudanica*. Therefore, the bwa mem aligner (or other) is appropriate for this analysis.

+ ****BLAST searches****: Several `blast` runs were performed but there is no information on parameters used in the Materials and Methods section for any of these runs. The authors should add this information.

We appreciate the reviewers suggestion here to include parameters of BLAST searches. We used the default parameters for all BLAST searches (BLASTn, BLASTp, tBLASTn, BLASTx). We have now included a section in the Supplementary Information Materials and Methods before the first instance of BLAST detailing the methods used.

- **Candidate genes**: The authors performed an extensive analysis of the two genomic regions linked to schistosome resistance. They identified interesting families of candidate genes and proposed working hypotheses for future work. However, this could be significantly strengthened by a small addition: Are these genes expressed in hemocytes, which are known to be the effector cells involved in resistance? I understand that, despite having performed RNA-seq on the *B. sudanica*, it is unlikely that the authors have tissue specific RNA data. However, there should be enough data on *B. glabrata* hemocytes to explore this (<https://doi.org/10.3389/fimmu.2022.956871>, <https://doi.org/10.1007/s00251-021-01236-3>, <https://doi.org/10.3389/fimmu.2018.02773><https://doi.org/10.1186/1471-2164-9-634>).

We thank the reviewer for this excellent suggestion. We do not have tissue specific expression data for *B. sudanica*. Following the reviewer's advice, we interrogated the *B. glabrata* hemocyte transcript sequencing (tissue specific or single cell RNA sequencing (scRNAseq)) datasets from three published studies (all of which were mentioned by the reviewer): Li *et al.* (2022), Pichon *et al.* (2022) and Dinguirard *et al.* (2018). We used BLASTp (using the input query as our *B. sudanica* SudRes1 and SudRes2 protein sequences) to identify the best matching orthologs in the *B. glabrata* BB02 genome used in these studies. In short, none of the top BLAST hits of the genes in *Sudres1* or *2* were reported as expressed in hemocytes. Thus, there are no clear orthologs of the *Sudres1* and *2* genes expressed in *B. glabrata* hemocytes.

However, many of the genes in these regions belong to large gene families, and thus the identification of true orthologs is somewhat challenging, and also might be complicated by the annotation of the genes in *B. glabrata* if not highly complete. In this same analysis, if we consider the top 5 BLAST hits (rather than the top one) of *B. glabrata* genes, we see a receptor-type tyrosine- phosphatase T-like protein (BGLB035129) from Pinchon *et al.* (2022) reported to be expressed in hemocytes that is the third best match to *B. sudanica* *SudRes1* gene BSUD.13443. The BGLB035129 gene in *B. glabrata* has tandem PTP domains, but not the MEGF or GBD domains we observed in *SudRes1* RPTPs, which as mentioned may be a result of the *B. glabrata* annotation not capturing the full gene or suggests that the genes are paralogs rather than direct orthologs. We also note that many (29 in total) genes described only as MEGF containing proteins in the *B. glabrata* annotations are present in the Li *t al.* (2022) and Pichon *et al.* (2022) single cell RNA-seq datasets. We have therefore added a sentence in the discussion in the section 'Evolutionary dynamics...' mentioning the presence of these genes in these hemocyte datasets.

Comments on the text

- **l. 30**": "snail immunity": There is no clear demonstration of the role of the genes under the QTLs in the snail immunity. The genes are good candidates by their features but a formal demonstration is needed to claim that these two QTLs are related to immunity.

We thank the reviewer for this important point. We changed “immunity” to “resistance”

- **l. 98-99**": "1.5x coverage per snail (Dataset S1)": 1.5x per snail seems low and may have prevented the identification of other important regions. Another, maybe more relevant metric could be the read depth of coding regions (where the causative loci are likely located) because the snail genome contains many repeats, which may lower the alignment efficiency and the average coverage/read depth.

1-2x coverage is a standard recommended coverage of pool-seq to maximize power to estimate allele frequencies for a given sequencing effort (e.g. Boitard et al. 2012 doi: 10.1093/molbev/mss090; Ferretti et al. 2013 doi: 10.1111/mec.12522; Tilk et al. 2019 doi: 10.1534/g3.119.400755). Median coverage at coding sites was only 1% higher than at noncoding sites.

- **l. 128**": "($p < 1e-14$)": A p-value alone is meaningless. The authors should at least provide the test used. Access to the code written for analyzing the data would have been a plus.

This was calculated with logistic regression, which we now state.

- **l. 138**": "GWAS ancestry signal is not caused by interspecies hybridization": This section is confusing. The use of *B. choanomphala* is sensible; however, the result of the admixture analysis contradicts the authors' claim. If *B. choanomphala* belongs to group A, then the group A has *B. choanomphala* ancestry and hybridization is ongoing, with group B representing the ancestral *B. sudanica* population. However, the analysis itself seems strange. My understanding is that it was first done with *B. sudanica* samples from the field only, which led to the identification of the two groups, and then the *B. choanomphala* and lab *B. sudanica* were assigned to these groups. However, this analysis should be performed using all samples at once to define populations and formally test for hybridization.

We now include a PCA (Fig S7) that assesses GWAS snails together with *B. choanomphala* and *B. sudanica*. This shows that the difference between Populations 1 and 2 in the GWAS snail ancestry is orthogonal to the difference between species.

In addition, the *B. choanomphala* sample in Fig. 2 is not apparent despite its mention in the caption.

This was an error in the caption. We did not include *B. choanomphala* in the ADMIXTURE analysis. *B. choanomphala* is only used in the PCA in Fig S7.

- **l. 280**": "immune related genes" should be "genes possibly related to immunity". The genes highlighted have not been demonstrated to be involved in immune reaction in the current system.

Changed the text as the reviewer suggested

In the Supplementary materials:

- **p. 4**": "We chose the model with the lowest AIC": what is the model chosen? The authors should add it.

We have added the model to Fig 2E.

Comments on the figures

- **Fig. S11**": How was the read depth normalized? Slight differences in sequencing depth between libraries could also explain the observation.

We have now normalized the depth in this figure (now Fig S14) and explained what we have done in the methods. The main point of this figure is that for both pools, depth at *SudRes1* (variable solid lines) is lower than the genome-wide average. However, we agree with the reviewer that normalizing the depths makes it easier to compare between pools, which indeed have different average depths, both genome-wide (due to differences in sequencing effort) and in this region (due to biological differences in duplications/deletions).

MINOR COMMENTS

=====

Comments on the text

- **l. 89 / 91**": "1400 F1" / "1,109 snails": the authors should be consistent in number notation. This should be check throughout the text.

We have changed "1400" to "1,400" and similarly checked number notation throughout the text, except when part of a name (e.g. BSUD.25704 or NovaSeq 6000).

- **l. 125**": "(Fig. S5)": This figure is mentioned before Fig. S2. Same with Fig. S6 (l. 138) and Fig. S7 (l. 140). The authors should reorder supplementary figure accordingly to the order of their appearance in the text.

We have reordered the supplementary figures based on the order of appearance in the text.

- **l. 264***: "confirmational" should be "conformational".

Changed as suggested.

- **l. 276-277***: "The *SudRes1* region was ... in *B. sudanica* (27)": At first read, this sentence appears misleading as it suggests that *SudRes1* was already identified in ref. 27. The authors should rephrase this to avoid confusion.

We see the confusion and have now edited it to read “Prior to its identification in this GWAS, the region now designated as *SudRes1* was previously noted as showing exceptionally high nucleotide diversity (27), which when coupled with the pooled-GWAS results suggest that pathogen-mediated balancing selection may act on these genes as previously hypothesized.”

- **l. 322***: "they do" should be "*B. choanomphala* does" if I understand correctly.

It appears that we have not communicated this portion of the manuscript very well given this misunderstanding along with similar comments above.

The sentence in question is “Third, while *B. sudanica* and the deep-water taxon *B. choanomphala* are closely related, perhaps being ecophenotypes (32), sympatric, and distinct in parasite susceptibility (47), they do not represent the ancestry groups and cluster with *B. sudanica* having a majority Population 1 ancestry.” We definitely mean that they do NOT represent the ancestry groups.

We have revised this sentence to say: “Third, this cryptic population structure does not align with divergence between *B. sudanica* and *B. choanomphala* which have been considered sister species or ecomorphs of a single species in Lake Victoria (32). Rather, Population 2 *B. sudanica* is distinguished by a set of alleles that do not appear to be common in either *B. sudanica* or *B. choanomphala*, and thus represents unique diversity that was not previously recognized.” We hope this, along with our figure changes, makes this concept more clear.

- **l. 335-346***: "The collection site ... lake areas": A map would be very appreciated to illustrate the text.

We have added a geographical map of this area (Fig S1).

- **l. 359***: "*a priori* candidates": I suspect the authors refer to the PTC1 and 2 loci. The authors should cite them explicitly for helping the audience unfamiliar with the system.

In Supplementary Data 3, the “Purpose” column, the a priori markers are indicated. These include *PTC1* and *PTC2* as well as candidates identified in *B. glabrata*. We apologize that this file may not have been previously viewable.

In the Supplementary materials:

- **p. 2**:

+ "timed" should be "trimmed".

Corrected

+ "MINLEN:50)" should be "MINLEN:50".

Corrected

+ "(GCA_000237925.5" should be "(GCA_000237925.5)".

Corrected

- **p. 3**:

"Schistosoma 16S amplicon": It is unclear if the authors refers to the 16S rRNA gene of the schistosome mitochondrion or if this is a typo and refers to the 18S rRNA gene in the schistosome nuclear genome.

Corrected – added “rRNA (mitochondrial)” after 16S in this sentence to avoid confusion.

- **p. 4**:

"if each allele has >10% the coverage of the other allele the genotype is designated heterozygote": This section is confusing to me. Are the authors considering the heterozygous state when allele A has 10 reads and allele B has 11 reads, or when allele A has 10 reads and allele B has 1 read? In the first case, what is the upper limit? In the second case, 10% seems very low to consider the individual heterozygous. Is there a rationale behind this threshold? In any case, the authors should clarify this section.

We have added Fig S4 to help clarify our strategy, and also added text to explain. All genotypes with less than 20 reads are counted as missing. Otherwise, if there are two alleles and the less-common allele has a coverage >10% of the more-common allele, it is counted as a heterozygote. This threshold was determined empirically, as the lines in Fig S4 (defined by this threshold) fall neatly between the clusters representing heterozygotes and homozygotes. There are very few genotypes at the 10% threshold, but there are many genotypes in the central “heterozygous cluster” with a skewed frequency between 10-50% for the less-common allele. So 10% seems like a reasonable cutoff. In the first example the reviewer suggests, (10 A reads, 11 B reads), we would call it a heterozygote. In the second example (10 A reads, 1 B read) it would be missing data. We need at least 20 reads, so 18 A reads and 2 B reads would be called a heterozygote, though this is an edge case and few genotypes are at such an extreme.

- **p. 7**:

"The dotted lines ... no correlation.": I do not understand this sentence. The authors should rephrase it.

We agree that the dotted lines were confusing. We have revised this figure (now Fig S2) in light of another reviewers' comments, and these dotted lines are now gone.

Comments on the figures

- **Fig. 1 & Fig. 3A**: characters from the y axis labels and the legend did not display correctly in Adobe Acrobat Reader v. 2024.005.20399.

With the resubmission, we have included Fig. 1 and Fig. 3 additionally as .png files so that reviewer can observe correct axis labels without formatting issues.

- **Fig. 2**: The authors should place the letter C on the left of the figure for consistency and readability.

Done.

- **Fig. S2**: The authors should change the color palette chosen for the linkage groups as this is already used for designated A and B groups in Fig. 2 and is therefore confusing.

We have changed this figure (now Figure S8) so that it is presented in black and gray.

REVIEWER COMMENTS

Reviewer #1 (Remarks to the Author):

The revised manuscript has improved significantly. However, several important concerns regarding the rigor of the study remain and should be addressed.

We appreciate the reviewer's contribution, and we have addressed these concerns as described below.

Both the population structure analysis and the genotype-phenotype association analysis should account for familial relatedness among samples. While the population structure analysis has been revised using appropriate methods, familial relatedness was not considered in the genotype-phenotype association analyses (Fig. 2B–E). A conservative approach would be to restrict the association analysis to unrelated individuals. Alternatively, methods that incorporate a genetic relatedness matrix (GRM) into regression models could be employed.

We thank the reviewer for pointing out the potential effects of relatedness that our previous analyses may not have accounted for fully. To address this, we have added another analysis in the supplemental information (pg 5) that describes how we used the method suggested by the reviewer—incorporating a relatedness matrix into the regression models. The results of this new analysis strongly support the loci that we previously identified through our original analyses, showing the robustness of our results (see new Figure S10). The analysis identified a third locus that now crosses the significance threshold. Because this locus was only identified in a single analysis, we mention the result, but we prefer to take a conservative approach and focus on the two loci that are revealed in all our analyses. The description of the analysis in the supplement is as follows:

"To assess whether close family relationships drive these results, we first used snpgdsIBDKING (16) to identify snail pairs with estimated KING-robust kinship coefficients. For each marker we then fit a linear mixed model against the infection phenotype using the Imm.diago function from the gaston v1.6 package (19), incorporating the eigen decomposition of the kinship matrix with the eigenK argument. As above, we ran two different analyses: one including our previously-defined ancestry estimate as a covariable, and one restricted to just ancestry group A or B. We calculated p values from the z-score obtained as $BLUP_beta/\sqrt{varbeta}$."

In addition, the revised submission includes supplementary data that were not part of the initial submission. The dataset numbering does not follow the order of appearance in the manuscript. Furthermore, the use of commas as thousands separators is inconsistent within and across datasets and should be standardized.

We thank the reviewer for the observation of supplementary dataset order of appearance. The dataset order now follows order of appearance in the manuscript, when also considering main Figure legends in the manuscript, for example, some citations for Dataset S2 are in the caption for Figure 1, which appears in the text before Dataset S2. We have

however changed the order of Dataset S5 and S6 as these had been incorrectly placed out of order following the manuscript edits.

We have now changed all numbers so that they appear with thousand separator comments.

Dataset S1

The number of reads mapping to *S. mansoni* is high, representing approximately 15-25% of total sequencing reads for both infection "negative" and "positive" pools. The proportions of *S. mansoni* reads also differ considerably between technical replicate pools. According to the methods section, *S. mansoni* reads were removed prior to mapping to the *B. sudanica* genome. However, the input read counts for *B. sudanica* alignment (Column AR) exceed the number of reads remaining unmapped to the *S. mansoni* genome (Column AC). Clarification is needed regarding this discrepancy. Moreover, there are multiple columns labelled "raw total sequences," the meaning of which is unclear.

The reviewer observed what they thought was a discrepancy in the number of reads that were used to align to the *B. sudanica* genome in Dataset S1 following our pipeline (remains as is detailed in Supplementary Information methods section) which includes aligning to the *S. mansoni* genome first, and removing reads that align to the *S. mansoni* genome confidently (based on CIGAR value being ≥ 70). This means that, following the read alignment to the *S. mansoni* genome, all reads that do not confidently align to *S. mansoni* genome (i.e. have a CIGAR value < 70) are still included in the data (now in column W in Dataset S1 with header: 'reads mapped and paired to *S. mansoni* genome (CIGAR value < 70 - not confidently aligned to *S. mansoni* genome)', which are also used in the alignment to the *B. sudanica* genome. We have now included a new column (column V: 'Number of reads removed (CIGAR value > 70 - aligned confidently to *S. mansoni*)', which displays the number of reads that were confidently aligned (CIGAR > 70) to the *S. mansoni* genome from each negative and positive sequencing pool (and technical replicates), and that were removed using the FilterMappedBamCigar perl script (this is detailed in the Supplementary Information methods). The proportion of reads that align confidently (CIGAR > 70) to *S. mansoni* in negative samples [0.7% i.e. $(13,586,368 + 10,294,946) / (1,904,115,680 + 1,414,223,590)$; see columns L and Q] is lower than those confidently aligned in positive samples [1.6% i.e. $(33,738,358 + 52,121,654) / (2,107,579,300 + 3,250,124,836)$]. We hope this description clarifies this to the reviewer, and also we hope the reviewer finds the updated table, and headers in Dataset S1 helpful to interpret the bioinformatic analysis performed, particularly in this part where *S. mansoni* reads are removed.

We have relabeled all columns to make clear what the 'total sequences' refer to.

Column A: No column header is provided.

Header ‘Sequencing pool’ now added.

Column B: SRA accession numbers would be preferable to file names.

Now added SRA accession number in column C.

Column C: The meaning of “Lane D” should be clarified.

This was a typo, it should have denoted ‘ID’, now corrected.

Column E: The "file size (bytes)" field should be removed.

Removed.

Columns F, K, O, U, AF, and AP: These columns appear redundant given the information provided in the header row.

Agreed and removed.

Column S: Some cells contain missing data.

Corrected, all columns now contain complete data.

Dataset S2

Clarification is required regarding the assignment of contig_1504 and contig_351 to linkage group 1B. Although these contigs are linked, neither could be mapped to BioGlab47 Chromosome 1.

Orthology of *B. sudanica* contigs to *B. glabrata* linkage groups has been previously described in Pennance et al. 2024, BMC Genomics. So even though the particular amplicons for these two markers do not show conclusive orthology to *B. glabrata*, we know that the contigs they are on match *B. glabrata* Linkage Group 1, based on other genes found on these contigs. We have added a column to Dataset S2 to show this.

Dataset S3

P values greater than 1 are present in Column K and should be corrected.

This was a formatting mistake on our part in which some values which should say “NA” were instead listed as integers. We have corrected the table.

The genotype missing rate for individual markers should be reported. Allele 1 and allele 2 nucleotides and their counts for each population used in the association testing should be provided. A separate tab or dataset should be included to show the genotype matrix for

individual *B. sudanica* samples (503 samples x 470 markers). This should include SRA accession numbers for each sample, along with relevant metadata such as ancestry proportions, infection status, and relatedness.

We have generated a new supplemental table, Dataset S4, containing this information. We agree this is a helpful addition. Generating this table was a useful exercise which revealed that we had switched the values of missingness per-locus and per-individual in the text, so we have corrected that.

Datasets S4 and S5

The Excel files require formatting revisions. Some cells are colored, bolded, or highlighted. If these formats convey specific information, explanations should be provided, otherwise, the formatting should be standardized. Additionally, certain rows lack gene IDs and gene coordinates (i.e., both fields are marked as NA). Furthermore, Dataset S4 contains two tabs with redundant information, which should be consolidated or clarified.

The colors/bolded/highlighted cells are carryover from our analysis, and do not relate to any specific information, so formatting now standardized. In the tables, NA refers to where orthologous genes were not identified (either outside of specific SudRes1 or SudRes2 regions, and therefore not searched for) or where orthologs could not be confidently established. This also goes for gene coordinates, where either start or end coordinates could not be confidently established from our orthologous gene searches. To the Table descriptions for Dataset S4 and S6 (now changed to S5 and S7 following previous comment and addition of new table) in the supplementary information file, we have now added description of what NA represents: “NA represents missing information where information for specific genes (i.e. *Biomphalaria* spp. gene orthologs or gene coordinates) were not identified in the current study.”

Comments on newly included or revised figures:

Figure S2

Significance testing results should be included.

We have fixed this.

Figures S3 and S5

Definitions or descriptions should be provided for the terms “true positive,” “true negative,” and “false negative.” Alternatively, these labels should be revised for consistency with the terminology used in the main text and other figures.

We now explain these terms in the “Snail phenotyping” section of SI Materials and Methods.

Reviewer #1 (Remarks on code availability):

They provided the commands used in the analysis with detailed annotation. There are several perl scripts used for data processing, each performing simple tasks.

Reviewer #2 (Remarks to the Author):

We appreciate the reviewer's contribution.

Reviewer #3 (Remarks to the Author):

The manuscript presents a genome-wide association study of African *Biomphalaria* snails, which serve as intermediate hosts for the parasite responsible for schistosomiasis in an endemic region of high transmission. I coincide with the authors that the recent revisions significantly enhance the quality of the manuscript without altering its primary conclusions. In fact, these modifications further corroborate the results indicating that the two genomic loci, *SudRes1* and *SudRes2*, are associated with resistance to *Schistosoma mansoni*.

I have examined the newly included PCA analysis and the Supplementary Information section titled "Validation of variants by amplicon sequencing." These additions provide greater clarity, transparency, and reproducibility to the ancestry analysis, which I find creditable.

Upon reviewing the updated GWAS methodology, I gained a clearer understanding of the strategies implemented to incorporate ancestry as a covariate. I agree that accounting for ancestry through multiple regression analyses appropriately isolates genetic markers that have a meaningful effect on the phenotype.

Regarding the section titled "Evidence of a shifting snail population structure in Lake Victoria could lead to increased infections," I verified that one of the publications I recommended has been duly cited in the discussion. I also appreciate the rationale for excluding Sire et al., recognizing its limited relevance in this context. In addition, I reviewed the revised Figure S7, which includes *B. choanomphala* in the ancestry-controlled PCA. The updated discussion more effectively addresses the observed cryptic population structure and the interspecies relationships among *Biomphalaria* snails in Lake Victoria.

The discussion of how ongoing admixture signatures may be influenced by the hydrologic patterns of Lake Victoria is now more nuanced and scientifically compelling. I agree with the authors that these findings highlight the substantial variation in pathogen resistance between closely related snail populations. This variability likely contributes to the persistent schistosome transmission in lake regions harboring highly susceptible *Biomphalaria* populations.

I also examined the new comparison in the discussion section between single-cell RNA sequencing datasets from *Biomphalaria glabrata* hemocytes and the study's findings. This interspecies comparison greatly enhances the contextualization of the results. I concur with the authors that incorporating this perspective enriches the study, while the observed substantial differences between *B. sudanica* and *B. glabrata* justify limiting an extensive review of the latter in this manuscript.

Furthermore, I confirmed that the figures now correctly display the axes, enhancing the accuracy of data presentation.

Overall, I approve of this revised version of the manuscript and have no further suggestions or corrections.

We appreciate the reviewer's contribution.

Reviewer #3 (Remarks on code availability):

I confirmed that the code associated with the bioinformatic analysis has been made publicly available in a repository. Upon reviewing the scripts, I found them to be clearly written and fully reproducible. Even the manually created scripts are completely shared, ensuring transparency. Researchers wishing to replicate the analysis should have no difficulty doing so. Additionally, I verified that the code is logically sound, as the bioinformatic tools and overall pipeline adhere to standard practices commonly used in genetic data analysis.

Reviewer #4 (Remarks to the Author):

This is the second review of the manuscript now entitled "Genes linked to schistosome resistance identified in a genome-wide association study of African snail vectors" (566873_0) by Pennance, Tennessen et al.

The authors adequately addressed the comments and implemented the necessary changes.

We appreciate the reviewer's contribution.

There are a few typos in the main text:

- **l. 320**": "identified" should be "identified" (i missing).
- **l. 321**": "Consistency" should be "Consistently" (l missing).

We have fixed these.

There is one typo in the supplementary methods:

- **p. 4**": "due their ambiguous" should be "due to their ambiguous".

We have fixed this.

Reviewer #4 (Remarks on code availability):

While a complete workflow would have been appreciated to facilitate the reproducibility, the code, comments and usage messages are enough to rerun the analysis.